# Geologic characterization of nonconformities using outcrop and core analogues: hydrologic implications for injection-induced seismicity

Elizabeth S. Petrie[1], Kelly K. Bradbury[2], Laura Cuccio[2], Kayla Smith[2], James P. Evans[2], John P. Ortiz[4,5], Kellie Kerner[3], Mark Person[3], and Peter Mozley[3]

[1]Western Colorado University, Geology Department, 1 Western Way, Gunnison, 81231, USA

[2]Utah State University, Geology Department, 4505 Old Main Hill, Logan, UT 84322-4505, USA

[3]New Mexico Institute of Mining and Technology, 801 Leroy Pl Socorro, NM 87801, USA

[4]Computational Earth Science Group, Los Alamos National Laboratory, Los Alamos, NM 87544, USA

[5]Johns Hopkins University, Department of Environmental Health and Engineering, 3400 N. Charles St., Baltimore, MD 21218, USA

*Correspondence to*: Elizabeth S. Petrie (epetrie@western.edu)

**Abstract.** The occurrence of induced earthquakes in crystalline rocks kilometres from deep wastewater injection wells poses questions about the influence nonconformity contacts have on the downward and lateral transmission of pore fluid pressure and poroelastic stresses. We hypothesize that structural and mineralogical heterogeneities at the sedimentary-crystalline rock nonconformity control the degree to which fluids, fluid pressure, and associated poroelastic stresses are transmitted over long distances across and along the nonconformity boundary. We examined the spatial distribution of physical and chemical heterogeneities in outcrops and core samples of the Great Unconformity in the midcontinent of the United States, capturing a range of tectonic settings and rock properties that we use to characterize the degree of past fluid communication and the potential for future communication. We identify three end-member nonconformity types that represent a range of properties that will influence direct fluid pressure transmission and poroelastic responses far from the injection site. These nonconformity types vary depending on whether the contact is sharp and minimally altered (Type 0), dominated by phyllosilicates (Type I), or secondary non-phyllosilicate mineralization (Type II). Our observations provide geologic constraints for modelling fluid migration and the associated pressure communication and poroelastic effects at large-scale disposal projects by providing relevant subsurface properties and much needed data regarding common alteration minerals that may interact readily with brines or reactive fluids.

## 1 Introduction

Deep wastewater injection near the nonconformity between the Phanerozoic sedimentary sequence and Proterozoic crystalline basement in the midcontinent United States (Sloss, 1963) is the primary means by which produced formation fluids are disposed of in Class II injection wells (Murray, 2015). Increased rates of seismicity in this region are associated with large volumes of wastewater injection (Ellsworth et al., 2015; Keranen et al., 2013; Nicholson and Wesson, 1990; Petersen et al., 2016; Zhang et al., 2013), reduction of friction on pre-existing faults, and pressure diffusion away from the injection point is controlled by the permeability structure of the rocks in the subsurface (Goebel and Brodsky, 2018; Yehya et al., 2018). Recent midcontinent seismicity nucleates on faults in crystalline rocks km's from injection sites (Keranen et al., 2014; Weingarten et al., 2015; Zhang et al., 2016), and spans timescales of months to years' post-injection, indicating that pore-fluid pressures and/or poroelastic loads are transmitted across or along the nonconformity or through connected fracture systems (including joints, faults, and veins), in the crystalline rocks (Ortiz et al., 2019). The depths of seismicity (up to 11 km) at some injection sites suggest that crystalline basement permeability is perhaps moderate to high ($10^{-16}$ to $10^{-14}$ m$^2$); (Zhang et al., 2016) and is dynamically increased by elevated fluid pressures (Rojstaczer, 2008). Observations presented in this paper are also relevant to Class VI injection wells used for geologic sequestration of $CO_2$, several of our analogue sites include deep reservoirs being evaluated for $CO_2$ sequestration (Leetaru et al., 2009; Leetaru and McBride, 2009; Plains CO2 Reduction (PCOR) Partnership, 2020; Thorleifson, 2008).

Numerical modelling of fluid flow and/or loading stresses associated with poroelastic effects across nonconformities indicate that: 1) the presence of a high-storativity, low-permeability basal seal reduces potential for basement induced earthquakes; 2) poroelastic effects can trigger seismicity far away from the injection location; 3) the presence of conductive faults, including those that cut the nonconformity and those that are isolated in the basement can provide direct fluid or fluid pressure pathways, and 4) permeable cross-nonconformity faults may exhibit high rates of seismicity (Chang and Segall, 2016; Goebel and Brodsky, 2018; Ortiz et al., 2019; Yehya et al., 2018; Zhang et al., 2013).

In this paper we summarize geologic observations made at the nonconformity zone, the altered rock volume surrounding the nonconformable contact. This zone varies in thickness and is defined by mineralogic and structural alteration of the protolith rocks surrounding the nonconformity. We characterize nonconformity zones associated with Precambrian granite, gabbro, gneiss, and schists, that are overlain by porous sedimentary rocks including sandstone and mixed carbonate-clastic sequences. Study site locations were chosen based on their distribution within the midcontinent region and the suite of lithologies present (Fig. 1). These analogues represent the diversity of the nonconformity in the United States midcontinent region and are analogues for deep fluid injection from produced waters (Class II) and sequestration of $CO_2$ (Class VI). At each site we document the lithology and structural features of the rocks on either side of the nonconformity to characterize the range of rock types associated with the contact and identify any evidence of past cross-contact fluid flow. We present data on the mineralogic and structural heterogeneities observed in outcrop and core, and these observations serve as proxies for variation

in mineral alteration and deformation surrounding subsurface nonconformity zones which may impact the future migration of fluids along and across the contact.

We find three end member types of nonconformity zones. These zones range from diffuse to sharp (Type 0), can be phyllosilicate rich (Type I), or dominated by non-phyllosilicate secondary minerals (Type II). Each contact type observed in this study has a range of mineralized textures and structural discontinuities. Due to weathering, deformation, diagenesis and fluid-rock interactions, the nonconformity zone may be hydraulically heterogeneous at the mm to 10's m scales and influence the migration of fluid and fluid pressures away from the injection well. Characterizing variations in rock properties at the nonconformity zone is critical for safe implementation of deep fluid injection, as the dimensions and hydraulic properties of the rocks in the nonconformity zones impact the subsurface flow regimes (Ortiz et al., 2019). The lithologic character of the nonconformity zone has implications for hydraulically connected regions by allowing direct fluid communication, changes in pore fluid pressure, and/or poroelastic loads. Because pressure diffusion and fluid migration depend on the permeability structure at a given location, our work can be used to improve hydrogeologic models that test the impact of lithologic changes and cross-nonconformity fractures on the transmission of pore fluids and/or poroelastic stress. We present results from hydrogeologic models based on observations of nonconformity zone characteristics, testing the impact various nonconformity zone types have on transmission of pore fluids.

## 2 Geologic setting

The North American craton, Laurentia, includes the Precambrian shields, the platforms and basins of the North American interior and the reactivated Cordilleran foreland of the southwestern United States (Fig. 1). The craton includes Archean blocks, the Yavapai-Mazatzal, and Grenville accretionary belts, and failed rifts (Hoffman, 1988; Marshak et al., 2017; Whitmeyer and Karlstrom, 2007). Precambrian exhumation produced erosional surfaces on the top of the crystalline basement which were buried by Phanerozioc clastic and marine sedimentary rocks (Marshak et al., 2017; Sloss, 1988). The nonconformities studied in this paper are located within the Superior craton, an Archean basement complex of granite-greenstone or higher grade equivalent overlain by erosional remnants of Early Proterozoic platform facies (Hoffman, 1988), the Yavapai-Mazatlal province, 1.76–1.65 Ga juvenile arc terrane that includes the Central Plains orogeny (Karlstrom and Humphreys, 1998; Sims, 1990; Sims and Peterman, 1986), the Grenville province, 1.3-1.0 Ga imbricate thrust slices formed during continent-continent collision (Rivers, 1997) and the Midcontinental Rift, ~1.1 Ga failed rift system, dominated by volcanic rocks and basin fill sedimentary rocks (Ojakangas et al., 2001) (Fig. 1).

### 2.1 Study areas

### 2.1.1 Outcrop locations

Exposed at Presque Isle and Hidden Beach along the southern shore of Lake Superior, Michigan the nonconformity is defined by Proterozoic Jacobsville Sandstone overlying early Proterozoic altered peridotite crystalline basement (Fig. 2). The geologic history of the serpentinized peridotite is not well constrained, it is thought that the peridotite was serpentinized between 1.80-1.1 Ga (Gair and Thaden, 1968). The overlying Jacobsville Sandstone is a dominantly fluvial sequence of feldspathic and quartzose sandstones (Malone et al., 2016), and at this study locality consists of a variably indurated pebble

to cobble conglomerate and a lenticular planar to cross-bedded light red quartz arenite (Fig. 2). The Presque Isle outcrops are analogues for geologic sequestration of $CO_2$ in the deep saline Jacobsville Sandstone reservoir (Leetaru and McBride, 2009). The nonconformity at Gallinas Canyon is exposed along a 4-km long section in the southern-most Sangre de Cristo Mountains, New Mexico (Figs. 1 & 3). The outcrop exposure consists of the crystalline rocks of the Yavapai province, highly deformed compositionally layered quartzofeldspathic gneiss, amphibolitic gneiss, felsite, biotite schist, and granitic

pegmatite (Lemen et al., 2015), overlain by the Devonian to Mississippian shallow marine transgressive sequence of carbonate and clastic rocks of the Espiritu Santo Formation. The Espiritu Santo Formation consists of primarily limestone and dolomitic limestone, with a basal conglomeratic sandstone known as the Del Padre Member (Baltz and Myers, 1999). The rocks are exposed within north-trending fault bounded blocks uplifted during the Neogene Laramide Orogeny (Baltz and Myers, 1999; Lessard and Bejnar, 1976). This location provides an analogue for the Raton Basin to the east where injection

in Class II wells has been linked to basement earthquakes that began in 2001 (Nakai et al., 2017; Rubinstein et al., 2014).

**2.1.2 Core samples**

The R.C. Taylor 1 core samples the Cambrian La Motte Formation sandstone and sheared Proterozoic granitoids in the Central Plains Orogen of the 1.6 Ga Yavapai-Mazatal Province (Marshak et al., 2017; Sims, 1990; Whitmeyer and Karlstrom, 2007) (Fig 1). The borehole was drilled adjacent to the Cambridge Arch and is associated with a series of northwest trending

transpressional faults of the Central Plains Orogen (Sims, 1990; Whitmeyer and Karlstrom, 2007) (Fig. 4). In this core, the arkosic La Motte Formation, regionally called the Reagan and Sawatch Sandstones, is a fine-grained, well-sorted glauconitic sandstone deposited during a transgression and is an analogue for Cambrian sandstones being evaluated for sequestration of $CO_2$ (Carr et al., 2005; Miller, 2012).

The CPC BD-139 core, recovered from the Michigan Basin samples the contact between the Cambrian Mount Simon

Sandstone and Precambrian altered granitoid gneiss of the Grenville Front Tectonic Zone (Fig. 1 & 5). The Precambrian crystalline rocks captured in this core are characterized as granitic to tonalitic gneiss (Easton and Carter, 1995) that form the basement of the Michigan Basin. The Michigan Basin is a thermally complex intracratonic basin situated over the lower peninsula of Michigan. Unexpectedly high levels of thermal maturity in the Paleozoic strata of the basin is thought to be attributed to elevated basal heat flow occurring up until Silurian time, as well as the prior existence of ~2 km of

Pennsylvanian and Permian strata that has since been eroded (Everham and Huntoon, 1999). The Mount Simon Sandstone

reservoir is a unit of deep wastewater injection in Oklahoma and it is also targeted for $CO_2$ storage (Barnes et al., 2009; Dewers et al., 2014; Leetaru et al., 2009; Liu et al., 2011).

The BO-1 core samples the lower Cambrian Mount Simon Sandstone overlying a Precambrian layered intrusive complex of altered gabbro and other mafic intrusions and felsic dikes (Fig. 6) (Smith et al., 2019). The crystalline basement rocks are part of the Northeast Iowa Intrusive Complex and are associated with the Midcontinent Rift System (Anderson, 2012). The Midcontinent Rift System extends from Kansas to Lake Superior and then southward through Michigan (Fig. 1). The geologic features associated with the Midcontinent Rift System, include axial basins filled with basalt and immature clastic rocks along with evidence of crustal extension (Ojakangas et al., 2001) . The BO-1 core is analogous to several geologic settings anticipated in the subsurface of the midcontinent region where lower Cambrian rocks directly overly Precambrian mafic igneous rocks of the Midcontinental Rift System along the Great Unconformity (Gilbert, 1962; Mossler, 1995). Northwest trending fault systems near the borehole were identified by magnetic lineaments and are likely part of the regional NW-SE Belle Plaine Fault Zone (Drenth et al., 2015). The Midcontinent Rift System is being studied for deep injection of CO2 (Abousif, 2015; Wickstrom et al., 2010).

## 3 Characterization of the nonconformity

Given the recognized importance of direct fluid transmission, variation in pressure, and poroelastic loads on induced seismicity (Chang and Segall, 2016; Ortiz et al., 2019; Yehya et al., 2018; Zhang et al., 2013), we provide an overview of rock properties observed at the nonconformity using integrated outcrop-based studies in Michigan and New Mexico, and analyses of core from Michigan, Minnesota and Nebraska (Fig. 1).

### 3.1 Methods

To describe the nonconformity zone in core and outcrop and document structures and mineralogy across the boundary we use a variety of micro- to meso-scale methods including lithological and structural logging of outcrop and core, optical thin-section petrography and x-ray diffraction (XRD) mineralogic studies, whole-rock x-ray fluorescence (XRF) elemental analysis, and gas or air permeability measurements, when possible. We evaluated fracture distribution in outcrop and core noting fracture types, presence of fault and/or shear zones and associated mineralogy. XRD and XRF was carried out at Utah State University (USU) X-ray analysis laboratory and XRD was carried out at Western Colorado University (WCU) petrography laboratory. At USU XRD analyses were done using a Panalytical X'Pert Pro X-ray Diffraction Spectrometer (40 mA and 45 kV) with monochromatic CuKα radiation utilizing X'Pert Highscore software for phase analysis. Whole-rock XRF analyses were conducted at the Washington State University Peter Hooper GeoAnalytical Lab using a Thermo-ARL automated X-ray fluorescence spectrometer. At WCU XRD analysis was done using a Brucker D8 X-ray Diffraction Spectrometer (40 mA and 45 kV) with monochromatic CuKα radiation utilizing DIFFRAC.SUITE software for phase analysis.

Twenty-five samples from BO-1 drillcore were selected for gas permeability testing through Schlumberger Rock Mechanics and Core Analysis Services. Profile permeability measurements were made in steady state conditions with a mini permeameter where gas is injected directly onto the core slab surface. The profile permeameter has a measurable permeability range of 0.1 millidarcies to 3 darcies (~$9.9\times10^{-17}$ m$^2$ to $3.0\times10^{-12}$ m$^2$).

To illustrate effects of a reduced permeability above the nonconformity on fluid migration we compare three hydrogeologic models of basal reservoir injection that consider continuous and discontinuous zones of altered low permeability rocks above the basement. We develop three-dimensional models to assess fluid migration along crystalline basement faults using MODFLOW, a public domain finite-difference groundwater flow code (Harbaugh and McDonald, 1996; Harbaugh et al., 2000) that solves the following groundwater equation:

$$\frac{\partial}{\partial x}\left(K_x \frac{\partial h}{\partial x}\right) + \frac{\partial}{\partial y}\left(K_y \frac{\partial h}{\partial y}\right) + \frac{\partial}{\partial z}\left(K_z \frac{\partial h}{\partial z}\right) = S_s \frac{\partial h}{\partial t} + Q(x,y,z,t) \qquad \text{Equation 1}$$

where $h$ is the hydraulic head [L], $K$ is the hydraulic conductivity tensor [L/T], $S_s$ is the specific storage [L$^{-1}$], $Q$ is the fluid injection source term, (i.e., injection well) [T$^{-1}$], and $t$ is time [T]. Equation 1 represents single-phase, constant-density groundwater flow in a three-dimensional Cartesian coordinate system. Hydraulic conductivity is a lumped parameter that includes the influence of fluid and medium properties and is defined as $K = kr_f\,g/m$, where $K$ is hydraulic conductivity, m/s; $k$

is intrinsic permeability, m$^2$; $r_f$ is fluid density, 997 kg m$^{-3}$ (water); $g$ is the acceleration due to gravity, 9.81 m s$^{-2}$; and $m$ is the dynamic viscosity of the fluid, 8.9 ´ 10$^{-4}$ kg/m/s. Multiple researchers have implemented similar groundwater flow models in MODFLOW to investigate pore pressure propagation associated with basal reservoir injection (Zhang et al, 2013; Zhang et al., 2016), and a more exhaustive description of cases and boundary conditions can be found in Ortiz et al (2019). Each of our model simulations includes a 100 m-thick basal reservoir ($3\times10^{-15}$ m$^2$) underlain by 9.9 km of relatively low permeability (k$_x$

= k$_z$ = $3\times10^{-17}$ m$^2$) crystalline basement rock. A 20 m-wide conduit-barrier fault (k$_z$/k$_x$=105; k$_z$ = $3 \times 10^{-10}$ m$^2$) is present in all simulations as is an injection well located 150 m from the fault zone. Wellhead pressures reached over 50 m excess hydraulic head after 4 days in response to 5000 m$^3$/day of continuous injection.

### 3.2 Results

### 3.2.1 Lake Superior, Michigan

Outcrops of the nonconformity between late Proterozoic Jacobsville Sandstone and early Proterozoic altered peridotite crystalline basement are exposed at Presque Isle and Hidden Beach along the southern shore of Lake Superior, Michigan (Fig. 2) (Lewan, 1972). At Presque Isle the topographic relief of the basement nonconformity varies by 2.5 m over a distance of 1100 m of outcrop (Cuccio, 2017), is locally cut by small-offset faults (30 cm of throw), and is composed of mineralized

conglomerate in direct contact with the underlying serpentinized peridotite or is transitionally interbedded with the overlying

sandstone (Fig. 2B). Where present, the conglomerate consists of sub-angular to rounded chalcedony, gneiss, and greenstone cobble clasts with fine-grained, poorly sorted, hematite cemented angular quartz grains.

At Hidden Beach, poorly consolidated basal conglomerates of the Jacobsville Sandstone are in contact with the Precambrian Compeau Creek Gneiss. The quartz arenite consists of fine-grained, angular, moderately sorted quartz with some feldspar. Distinctive bleached open-mode fractures or reduction spots are associated with the lower Jacobsville Sandstone and range in orientation from near-vertical to bedding-parallel. The near-vertical reduction fractures (Fig. 2C) are not observed to extend into the basement.

Optical petrography across the transition from red sandstone protolith to a bleached fracture zone at Hidden Beach reveals a reduction in hematite grain coatings and cements. Whole-rock XRF analysis of the bleached areas of Jacobsville Sandstone indicates a minor depletion of $K_2O$, and a minor enrichment of $FeO$ and $MgO$, relative to the unaltered Jacobsville Sandstone (Fig. 7). At Presque Isle, mineral alteration products in the conglomerate include nontronite, with trace zeolites and iron oxides (Fig. 7). The underlying serpentinized peridotite is black to brown, with abundant white carbonate mesh veinlets and localized stockwork jasperoid veins up to 10 cm wide (Fig. 2). Jasperoid mineralization occurs along a few small faults that cross the nonconformity (Cuccio, 2017).

### 3.2.2 Gallinas Canyon, New Mexico

Devonian to Mississippian carbonate and clastic rocks of the Espiritu Santo Formation deposited on the Proterozoic quartzo-feldspathic and amphibolitic gneiss, biotite schist, and granitic pegmatite (Lemen et al., 2015) are exposed along a 4-km long section in Gallinas Canyon, eastern Sangre de Cristo Mountains, New Mexico. The nonconformity is cut by cm- to m's -displacement faults, at this location we characterize both the faulted and un-faulted nonconformity zone (Hesseltine, 2019; Kerner, 2015). Top of basement is defined by phyllosilicate rich zone with variable thickness, 0 to >5 m, that is truncated by the Del Padre Sandstone. Locally the Del Padre Sandstone is laterally discontinuous (Hesseltine, 2019) but is reported to be up to 15 m thick filling depressions in underlying crystalline rock elsewhere in New Mexico (Armstrong & Mamet, 1974). The carbonate and clastic rocks of the Espiritu Santo Formation include: 1-m thick massive, fine-grained, rounded to sub-rounded sandstone with calcite nodules, ~1- m of microcrystalline dolomite that transitions upward into a chert nodule limestone, interbedded mudstone and limestone and a massive microcrystalline limestone bed. A phyllosilicate-rich zone directly below the nonconformity is approximately 60-cm thick and is a poorly lithified zone that marks the transition from highly altered (weathering and hydrothermal alteration) to minimally altered crystalline rock (Fig. 8). The Precambrian crystalline rocks are cut by large thrust faults and smaller scale normal faults (Baltz and Myers, 1999; Lessard and Bejnar, 1976) with some faults juxtaposing sedimentary and crystalline rock (Fig. 8D).

The predominant lithology of the crystalline basement is gneiss, with minor schist, pegmatitic granite, and basalt. Mineral alteration is greatest directly below the nonconformity. This zone is enriched in sericite within feldspars, and clay minerals (mixed with hematite and associated with replacement of micas) (Fig. 8). Where cut by faults the nonconformity-associated

phyllosilicates form a matrix that surrounds more rigid grains such as quartz, suggesting deformation in this unit was accommodated by granular flow, a process associated with high pore-fluid pressure (Paterson, 2012). Microscopic fracturing has occurred within the crystalline basement, these fractures are mineralized with iron oxide, sericite, chert, and calcite. The majority of fractures within the crystalline basement occur along weak grains such as sericitized feldspar and altered mica or cut across quartz and feldspar grains. Authigenic calcite is rare within the crystalline basement, though commonly occurs as coarsely crystalline calcite cement within grain fractures in feldspar and sericitized feldspar.

Where faults cut the altered crystalline basement locally cataclasites are found throughout the fault core. Where faulted, the sedimentary rock damage zone includes large twinned calcite grains in fracture-filling cements, and cataclasites that lie along the edges of the calcite veins. The cataclasites include: pulverized quartz and feldspar grains, chert, pulverized protolith, as well as clay- and iron oxide-rich minerals. Quantitative microprobe analyses of the carbonate and fine-grained matrix composition within the sedimentary and basement fault cores reveals that all calcite vein elemental values have a slightly reduced level of Fe and Mg substitution for Ca than the calcite matrix. The fine-grained matrix within the sedimentary fault core is nearly pure silica, whereas the fine-grained matrix within the crystalline basement fault core is aluminium-rich (Fig. 8).

### 3.2.3 R.C. Taylor 1 Core, Nebraska

Core from the R.C. Taylor 1 wildcat well was obtained in 1953 in south-central Nebraska. We examined 19.2 m of core recovered over the Cambrian La Motte Formation sandstone and sheared Proterozoic granitoids in the Central Plains Orogen of the Yavapai Province (Marshak et al., 2017; Sims, 1990; Whitmeyer and Karlstrom, 2007). The arkosic La Motte Formation, regionally called the Reagan and Sawatch Sandstones, is a fine-grained, well-sorted glauconitic sandstone (Fig. 6).

The basal La Motte Formation and upper-most basement is cut by quartz, calcite, dolomite, and iron-oxide veinlets (Fig. 9). Iron-oxide veins cut quartz veins, and both are cut by calcite veins, providing evidence for fracture reactivation (Fig. 9). Below the La Motte Formation is a phyllosilicate-rich zone composed of 40 cm thick highly altered basement shear zone that overlies a minimally altered basement shear zone, both are comprised of fine crystalline sericitized feldspar and chlorite-rich shear zones, and overlie the coarse-crystalline, minimally altered granitic basement containing some sericitized feldspar (Fig.9).

The altered basement shear zone is composed of quartz, feldspar, biotite, chlorite, and dolomite (Fig. 9). Quartz and feldspars are disintegrated, well-developed chlorite, hematite and magnetite are altered from biotite, and granular disintegration has resulted in clay development. Open pore-space occurs between host-rock grains and neoformed clays. The basement shear zone is characterized by feldspar, quartz, mica, and the alteration minerals chlorite and dolomite (Fig. 9). The shear zones contain chlorite lined slip surfaces, and S-C fabrics within chloritizied zones. The shear fabrics are cut by open-mode quartz, sparry calcite, iron-oxide and dolomite veins. The basal, moderately altered basement unit is a coarse-crystalline granite composed of feldspar, quartz, biotite, and hornblende (Fig. 9). Chlorite is present and associated with minor shear fabrics. Open-mode calcite, dolomite, and quartz veins parallel and cross-cut the chlorite-rich shear fabrics and cut quartz and feldspar

crystals (Fig. 9). In the coarse-crystalline granite altered feldspars contain sericite that has formed adjacent to twin planes. Open-mode fractures mineralized with dolomite, calcite, and hematite occur in the lower 7 m of the La Motte Formation and are observed through the underlying granitic shear zone covering 12.5 m of core.

### 3.2.4 CPC BD-139 Core, Michigan

The CPC BD-139 core, obtained in 1964 for the design of a brine disposal well, samples the contact between the Cambrian Mount Simon Sandstone and Precambrian altered granitoid gneiss of the Grenville Front Tectonic Zone. We divide the CPC BD-139 core into three lithologic units: a laminated sandstone, a finely foliated gneiss, and a gneiss with sub-horizontal white veins.

Sandstone grains are rounded to sub-rounded and moderately to well-sorted. The Cambrian Mt. Simon Sandstone in the Michigan Basin is characterized as a porous (5-15% pore space) arenite to sub-arkosic sandstone (Leeper et al., 2012). Permeability in the basal Mt. Simon Sandstone is reported to be between $1 \times 10\text{-}16$ $m^2$ to $1 \times 10\text{-}12$ $m^2$ (Frailey et al., 2011).

A discrete boundary separates the Mount Simon Sandstone from the underlying altered granitoid gneiss (Fig. 10). The uppermost 30 cm of the basement is composed of tan, fine-grained, dolomite horizon which grades into a dark green foliated gneiss cut by pink sub-vertical fractures over a span of ~5 cm (Fig. 10). The basal meter of the Mount Simon Sandstone is a tan, finely laminated arenite with minor amounts of iron-rich clay. The quartzo-feldspathic granitoid gneiss near the contact contains the following alteration products: zeolites, vermiculite, and Fe-, Mn-oxides, and carbonates including dolomite (Fig. 10). Dolomitization of the basement host rock re-appears ~2 meters below the nonconformity. The original basement foliation is preserved and is associated with micrometer-scale crystalline dolomite grains, radiating silica crystals, and sub-horizontal calcite and dolomite open-mode veins (Fig. 10). Trace amounts of ankerite, clinochlore, and vermiculite are also present in the dolomitized basement rocks (Fig. 10).

### 3.2.4 BO-1 Core, Minnesota

The BO-1 core was originally collected in 1962 as part of an exploratory mining project in Fillmore County, southeast Minnesota (Gilbert, 1962).This core provides a continuous 300 m section of altered and mineralized rocks of lower Cambrian Mount Simon Sandstone overlying a Precambrian layered intrusive complex of altered and unaltered gabbro, other mafic intrusions and felsic dikes (Smith et al., 2019; Fig. 11).

Sedimentary sequences in BO-1 extend to ~ 1.2 km where the nonconformity is marked by an ~ 12 cm zone of pervasive leaching and iron-hydroxide staining (goethite). Intense alteration extends into the basement rocks for ~ 21 m, with ~ 50 m of argillitic and propylitic alteration, and/or fracture mineralization observed to ~ 402 m depth (Fig. 11). Localized faults, hybrid and open-mode fracture surfaces intersect the sampled basement core from within ~ 1 cm of the nonconformity contact and extend to 475.5 m, fracture density decreases with depth (Fig 11). Slip surfaces exhibit oblique to dip-slip slickenlines and

range from mm's to cm's thick and are either coated in clay or contain mineral infillings (±carbonate, ±silica, ±chlorite, ±iron-oxides).

The Mount Simon Sandstone contains a ~ 0.5-meter zone of intense iron-hydroxide (goethite) alteration at the nonconformity (Fig. 11). This iron-hydroxide oxidized zone extends for several m's into the slightly altered and metamorphosed crystalline
basement rock. From petrographic and XRD analyses, we identify mineralogical assemblages (dolomite, siderite, iron-oxides, iron-hydroxides, illite, smectite, kaolinite-serpentinite, vermiculite) and textures that are indicative of weathering, diagenesis, and multiple episodes of fluid-rock interactions coupled with deformation within the broad ~ 50 m zone of intense alteration also marked by abundant structural discontinuities (fractures, faults, veins) across the nonconformity zone (Fig. 11).

Measured gas permeability values are highest above the nonconformity within the porous Mount Simon sedimentary reservoir
(up to 1000 millidarcy) and vary significantly from 0-500 millidarcy below the nonconformity contact. Locally permeability increases in direct correlation to the presence of structural discontinuities (Fig. 6).

### 3.3 Hydrogeologic models

The first model, (Fig. 12A), is a Type 0 nonconformity, represented by a sharp contact between basement and overlying injection reservoir. In the second model simulation, a Type I nonconformity, includes a 20-m-thick, low-permeability
$(k_x=k_z=3\times10^{-18}$ m$^2)$ zone (Fig. 12B); this layer is 1 order of magnitude less permeable than the basement host rock and a further 1 order of magnitude less permeable in the fault core. The continuous low permeability zone reduces the permeability of the basement fault damage zone by 4 orders of magnitude, making the fault damage zone non-conductive. Pressure does not propagate into the crystalline basement although there was some diffusion of the 2-m excess hydraulic head front to depths ≤ 500 m. In the third simulation, a discontinuous low permeability zone is present (Fig. 12C). Where this zone is absent, the
pressure front propagates into the basement along the fault damage zone to a depth of 2.5 km. The fault zone was not blocked by the low permeability zone, and elevated pore pressures propagated downward to depths of 2.5 km via the fault zone (Fig. 12C). Elevated fluid pressures likewise appeared to be forced down other areas where the low permeability zone pinches out, such as towards the right-hand side of Fig. 12C.

### 4 Discussion

The nonconformities examined in this study range from sharp contacts to zones several m's thick and exhibit a range of mineralized textures and structural discontinuities (Supplementary Table 1). We observe mineralogic alterations across the nonconformity that are expected to impact diffusivity and storativity, and the sites evaluated provide geological and hydrogeological analogues that aid in understanding the impact circulating fluids may have on altering rock properties at depth (Oliver et al., 2006). Based on observations made in this work, we divide nonconformities into three end-member types, (Table
1), Type 0 – a sharp contact between sedimentary strata and basement rocks; Type I – an interface dominated by phyllosilicates;

and Type II – an interface dominated by non-phyllosilicate secondary mineralization. All the nonconformity types observed in this study are cut by structural discontinuities, therefore several possible contact sub-types exist within these 3 proposed end members (Fig. 13). Based on our observations structural and mineralogical heterogeneities at the sedimentary-crystalline rock nonconformity are thought to control the degree to which fluids, fluid pressure, and associated poroelastic stresses are

transmitted over long distances across and along the nonconformity boundary. The structural elements and fluid-related alteration patterns observed in these analogue sites support the hypothesis that the nonconformity interface zone influences or controls the potential for cross-contact fluid flow and distribution of fluids within the crust.

Our collective field and core observations in various basement tectonic settings document the occurrence of significant variations in altered or mineralized zones that lead to contrasts in permeability across the nonconformity. Where present the

structural discontinuities includes small offset faults, shear fractures, and veins. In thin-section we note evidence for dissolution, recrystallization, new mineral growth, and veins that reflect mineralization or deformation at depth and are not the result of alteration due to weathering alone. Crack-seal textures and calcite twinning lamella, suggest vein mineralization at depth (Burkhard, 1993), and reactivation of pre-existing fractures document episodic fracture growth (Davatzes and Hickman, 2005; Laubach et al., 2004).

At a Type 0 nonconformity, the nonconformity zone is expected to prevent direct fluid pressure communication across the contact due to a significant contrast in rock permeabilities that would hinder cross-contact fluid migration while promoting migration parallel to the contact distributing fluids laterally away from the injection site (Fig. 13A); at a Type I nonconformity, a phyllosilicate dominated contact is expected to inhibit fracture propagation across the nonconformity (Ferrill et al., 2012; Larsen et al., 2010; Schöpfer et al., 2006) and therefore maintain a significant permeability contrast preventing direct fluid

migration. In such cases, nonconformities result in a poor hydrologic connection between the sedimentary section and deeper basement rocks (13B).

However, repeated brittle failure and mineralization, observed in Type I nonconformities, suggest that phyllosilicate dominated shear zones can act as a zone of mechanical weakness that can be reactivated allowing for development of fracture permeability. In this fractured nonconformity we observed alteration as deep as 5 m below the nonconformity in the crystalline

rocks examined, however, previous work highlights the potential for fractures and connectivity to basement fault zones at much greater depths (Duffin et al., 1989). Pre-existing basement shear zones that are reactivated may allow future fluid circulation during injection scenarios.

Type II nonconformities (Table1, Fig. 13C, Supplemental Table 1) are mineralized contacts, that include secondary alteration minerals found within 10 cm to several m's below the nonconformity. The mineralization due to fluid-rock interactions at the

Type II nonconformities suggests that deep fluid circulation occurs even without enhanced permeability from fractures (Cuccio, 2017) (Fig. 12C). This nonconformity type could prevent brittle deformation but may be more influenced by poroelastic loads. The impact of these contacts on hydrogeologic properties is not yet well understood or modelled.

The impact the morphology of the nonconformity has on the downward propagation of fluid pressures into the crystalline basement has been shown by several numerical hydrogeologic studies (Ortiz et al., 2019; Segall and Lu, 2015; Yehya et al.,

2018; Zhang et al., 2016). These models suggest that direct pore-fluid pressure communication (Ortiz et al., 2019; Segall and

250 Lu, 2015; Yehya et al., 2018) and significant changes in poroelastic stress (Goebel and Brodsky, 2018; Zhang et al., 2016) can occur well way from the injection zones. Numerical models predict that nonconformities with through-going fractures distribute fluid deeper into the basement rocks and that direct pore pressure communication can destabilize faults at depth (Ortiz et al., 2019; Segall and Lu, 2015; Yehya et al., 2018). All the nonconformity types observed here are cut by

structural discontinuities, and several possible contact sub-types exist within these 3 proposed end member scenarios (Fig. 12). Fractures, and especially fault zones, are expected to distribute fluids and propagate fluid pressures to a greater depth regardless of nonconformity type (Yehya et al., 2018). Because nonconformity interface zones with pre-existing deformation fabrics may be preferential flow pathways that distribute fluid pressure away from the injection zone, high-permeability damage zones transmit fluid pressure to greater depths than non-conduit fault zones (Yehya et al., 2018).

To illustrate effects of a reduced permeability above the nonconformity and the impact of permeable fault zones on fluid migration we compare three models of basal reservoir injection that consider continuous and discontinuous zones of altered low permeability rocks above the basement (Fig. 12). The Type 0 nonconformity, represented by a sharp contact between basement and overlying injection reservoir, results in lateral migration away from the injection well and downward migration where it encounters a fault zone (12A). In the second model simulation, a Type I nonconformity, the presence of a

20-m-thick, low-permeability zone and no connection between basement and sedimentary fault zones results in lateral migration and pressure does not propagate into the crystalline basement (Fig. 12B). In the third simulation, models a Type I nonconformity with a discontinuous low permeability zone, where this zone is absent, the pressure front propagates into the basement along the connected fault damage zone to a depth of 2.5 km and elevated fluid pressures appear to be forced downward were the low permeability zone pinches out (Fig. 13C).

Our collective field and core observations document the occurrence of significant variations in altered or mineralized zones which would impact permeability values associated with the nonconformity zone, and that alteration coupled with abundant structural discontinuities can result in relatively higher permeability that extends for 10's of meter's both into the crystalline basement rock below the nonconformity.

**5 Conclusions**

We define key rock types and structural elements of the nonconformity zone and split the analogue nonconformities into three end-member types. The three non-conformity end member types provide a broad hierarchy of nonconformities in the midcontinent (Supplemental Table 1) and are observed at nonconformity sites elsewhere. We expect these nonconformity types to either distribute fluid pressure away from the injection point or provide direct communication with basement rocks, moving fluids to a greater depth across the nonconformity. We observe that fractures cut all nonconformity types and expect

in these cases changes in fluid pressure or poroelastic loads could result in triggered earthquakes within basement rocks (Chang and Segall, 2016; Zhang et al., 2013). Numerical modelling of Type 0 and Type I end members that include fault zones predict downward propagation of fluid pressure and changes to poroelastic loads. The data presented here can be used to improve

model inputs for evaluation of cross-contact fluid and pressure communication whether through creation or modification of existing permeability or poroelastic pathways or through rheological changes associated with fluid-rock interactions. We show that conditions along the nonconformity zone vary, and the data from outcrop and core observations also suggest that injection of brines at depth may drive mineralogical alteration and potential fault zone weakening, these data can also be used to understand the impact that long-term storage of chemically reactive fluids has on rock properties (Callahan et al., 2020). Once fluids penetrate the basement, flow is likely controlled by fracture and fault systems and reactivation of pre-existing structures is possible. However, micro-porosity within basement rocks may enhance mineralogical changes over the long term and transmit fluids deeper in the basement while promoting short-term lateral migration along the nonconformity.

Our observations illustrate that the contact should not be treated as an impermeable barrier to fluid flow nor as one cut by faults of various permeabilities but should instead be evaluated on a site by site basis prior to injection of large fluid volumes.

## 6 Acknowledgements

This work was supported by collaborative U. S. Geological Survey NEHRP grants #G15AP00080 and #G15AP00081 awarded to Evans, Bradbury, Person, and Mozley, a Western Colorado University Professional Activity Fund grant to Petrie, and a USGS-USU cooperative agreement #G17AC00345 to Bradbury and Evans. Additional student support obtained from student research grants from the Geological Society of America (GSA) and JS Williams (USU Geosciences) grants awarded to Cuccio and Smith, the GSA Stephen E. Laubach Structural Diagenesis Award to Smith, a GDL Foundation grant, and Institute of Lake Superior Geology grant to Cuccio.

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

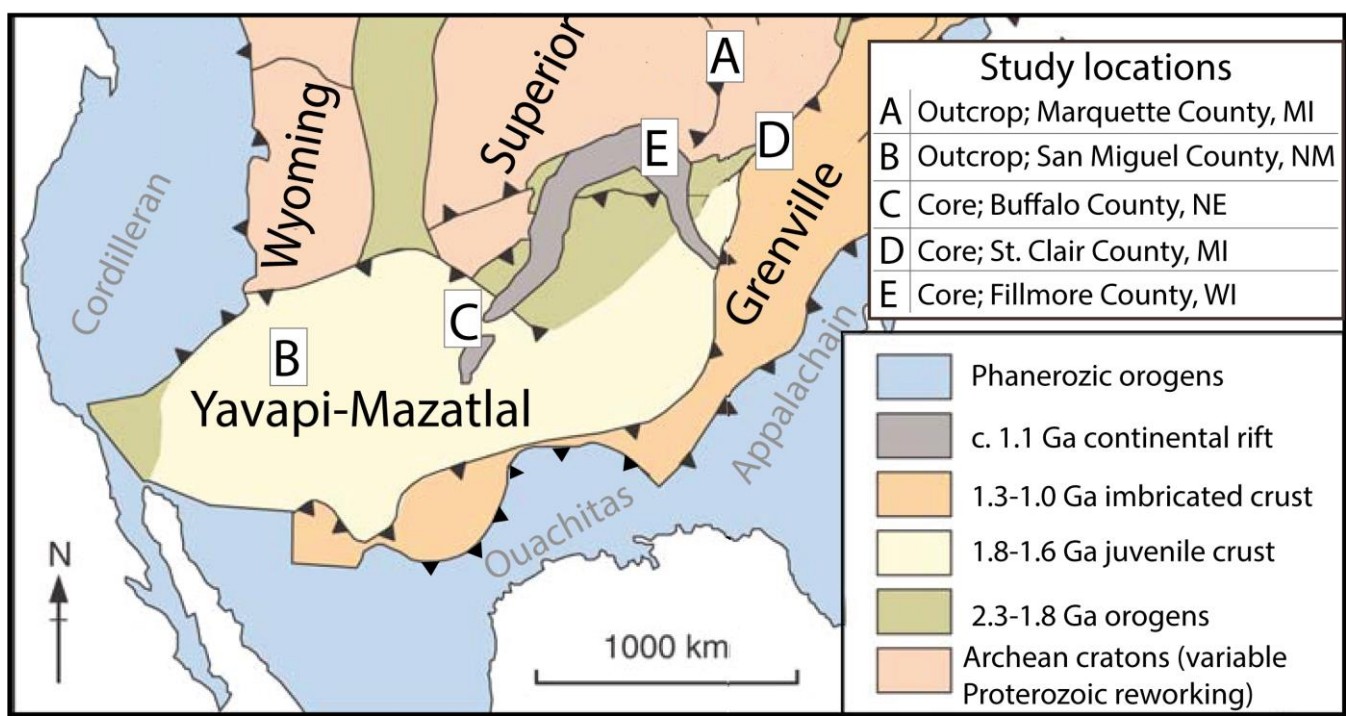

**Figure 1. Precambrian tectonic elements map with location of the nonconformity analogue study sites (after St. Onge et al., 2009).**

**A) Lake Superior, Presque Isle, Michigan outcrop, B) Gallinas Canyon, New Mexico outcrop, C) R.C. Taylor 1 core, D) CPC BD-139 core, and E) BO-1 core.**

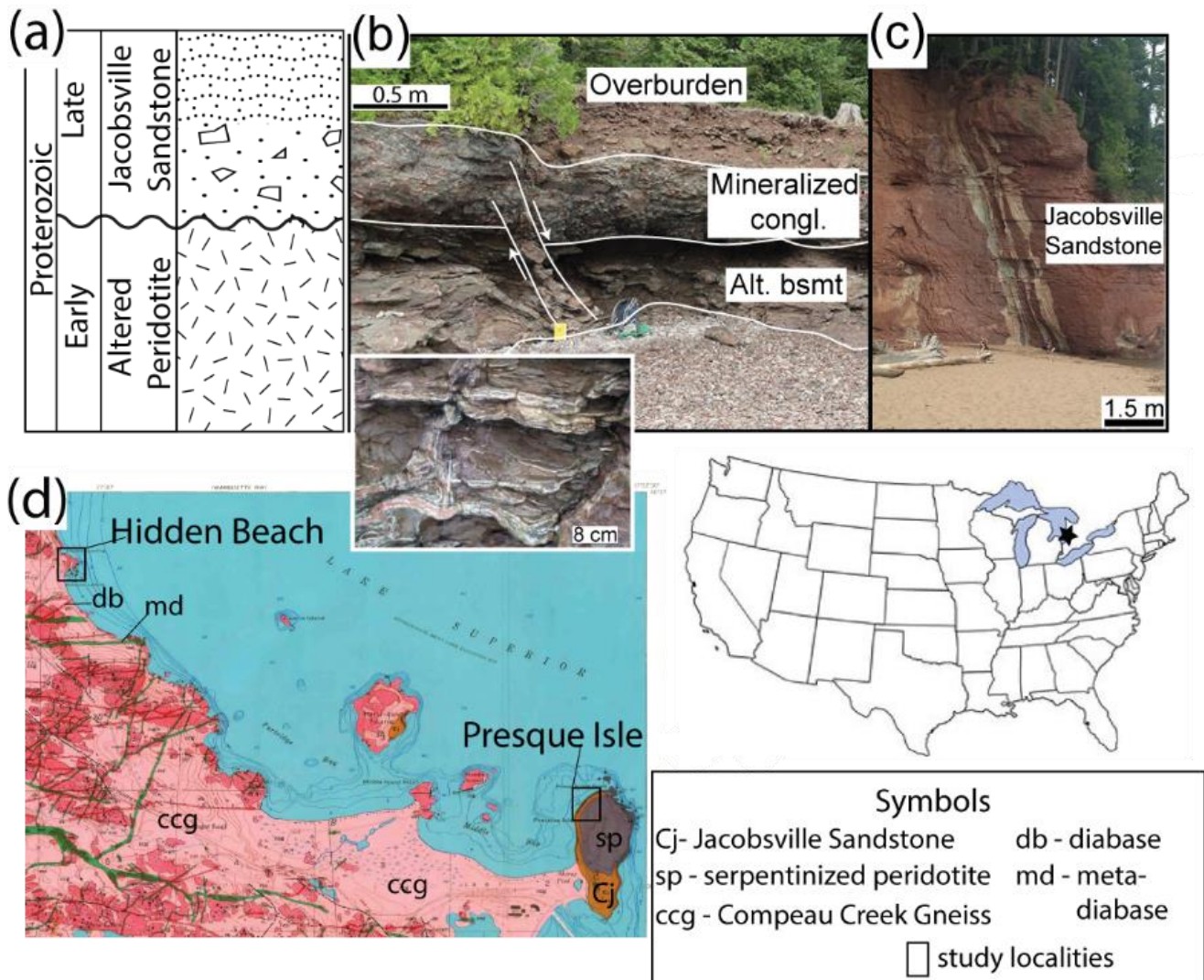


**Figure 2. A) Schematic lithologic log at Lake Superior Michigan where altered peridotite is overlain by the Jacobsville Sandstone at Presque Isle and mineralized conglomerates of the Jacobsville Sandstone overlie the Compue Gniess at Hidden Beach. Outcrop photos B) Presque Isle, outcrop of small fault cutting the contact between mineralized conglomerate of the Jacobsville Sandstone and the underlying altered basement, inset shows stockwork jasperoid veins in the underlying serpentinized peridotite basement**

**and C) Hidden Beach outcrops. At this locality the Jacobsville Sandstone overlies the Proterozoic altered peridotite basement rocks. D) Geologic map of the Marquette, Michigan field area, showing locations of Hidden Beach and Presque Isle, modified from Gair and Thaden, 1968.**

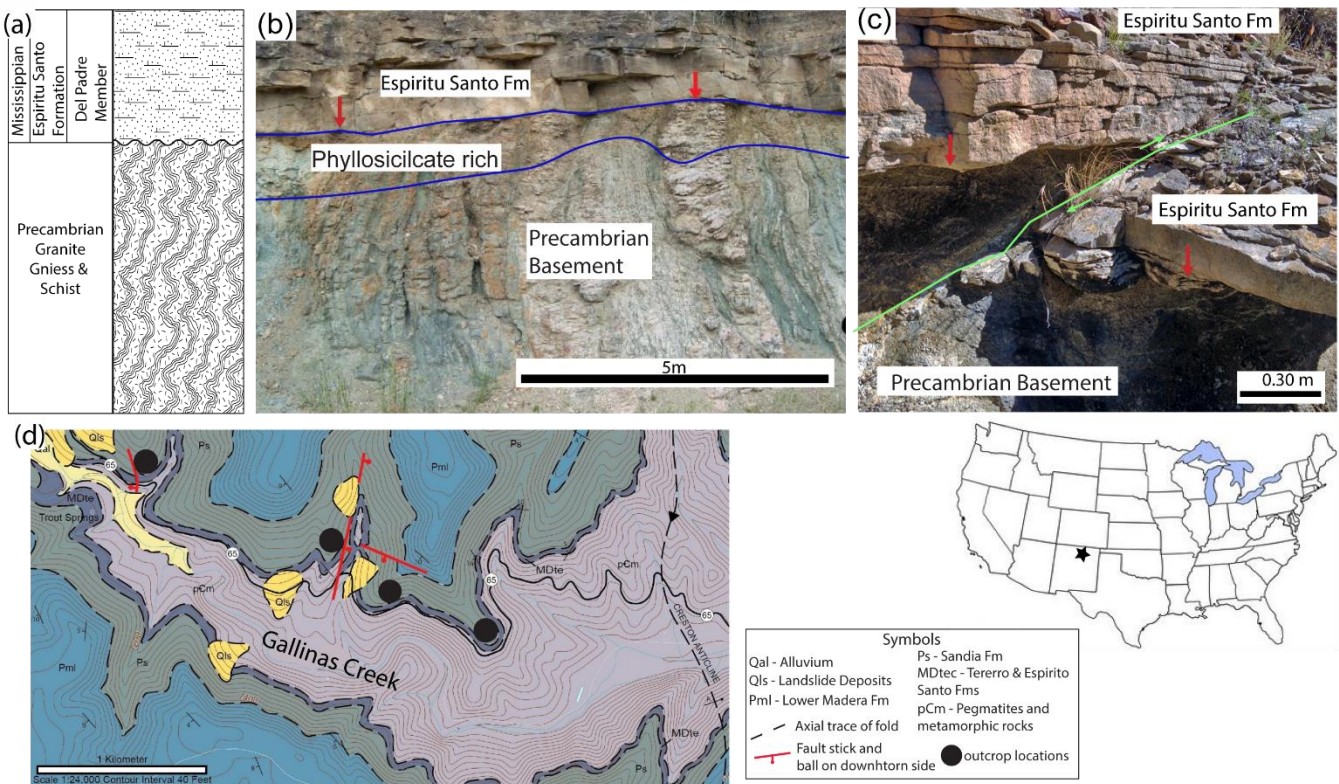

**Figure 3. A)** Gallinas Canyon, New Mexico outcrop lithology log. **B)** Precambrian granitic gneiss and schist is overlain by the
Mississippian Espiritu Santo Formation, red arrows mark the nonconformity, blue lines mark boundary of phyllosilicate
alteration. **C)** The 4 km long exposure in Gallinas Canyon, the nonconformity is cut by several cm- to m- displacement faults, red
arrows mark the nonconformity, fault shown by green line. **D)** Geologic map of Gallinas Canyon study area, modified from
Hesseltine, 2019. Faults that cut the nonconformity are shown in red with ball on down dropped, at map scale the nonconformity is
relatively planar, and parallels topographic contour lines.

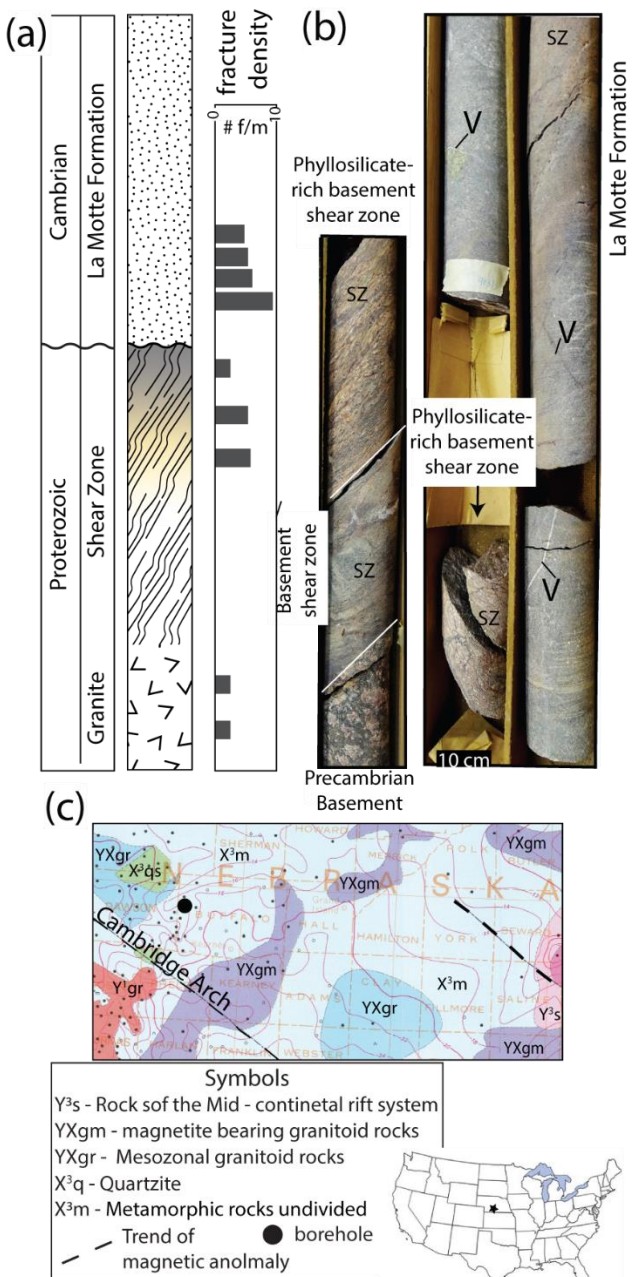


**Figure 4. A) Lithologic log of the R.C. Taylor 1 core, Nebraska, core from 3984-4038' (1214-1231 m) measured depth. The nonconformity occurs at 4018' MD (1225 m). Fracture density in core is based on number of fractures per meter of core. Four lithologic units are identified in the core, including sandstone, sedimentary rock hosted shear zone, altered basement shear zone, minimally altered basement. The nonconformity occurs between the altered basement shear zone and overlying sandstone of the La**

**Motte Formation. B) Photographs of the R.C. Taylor 1 core. Both the basement shear zone (SZ) and overlying sandstone are cut by veins (V) of quartz, calcite and Fe-oxides. C) Borehole location shown on the Precambrian basement map from Sims (1990).**

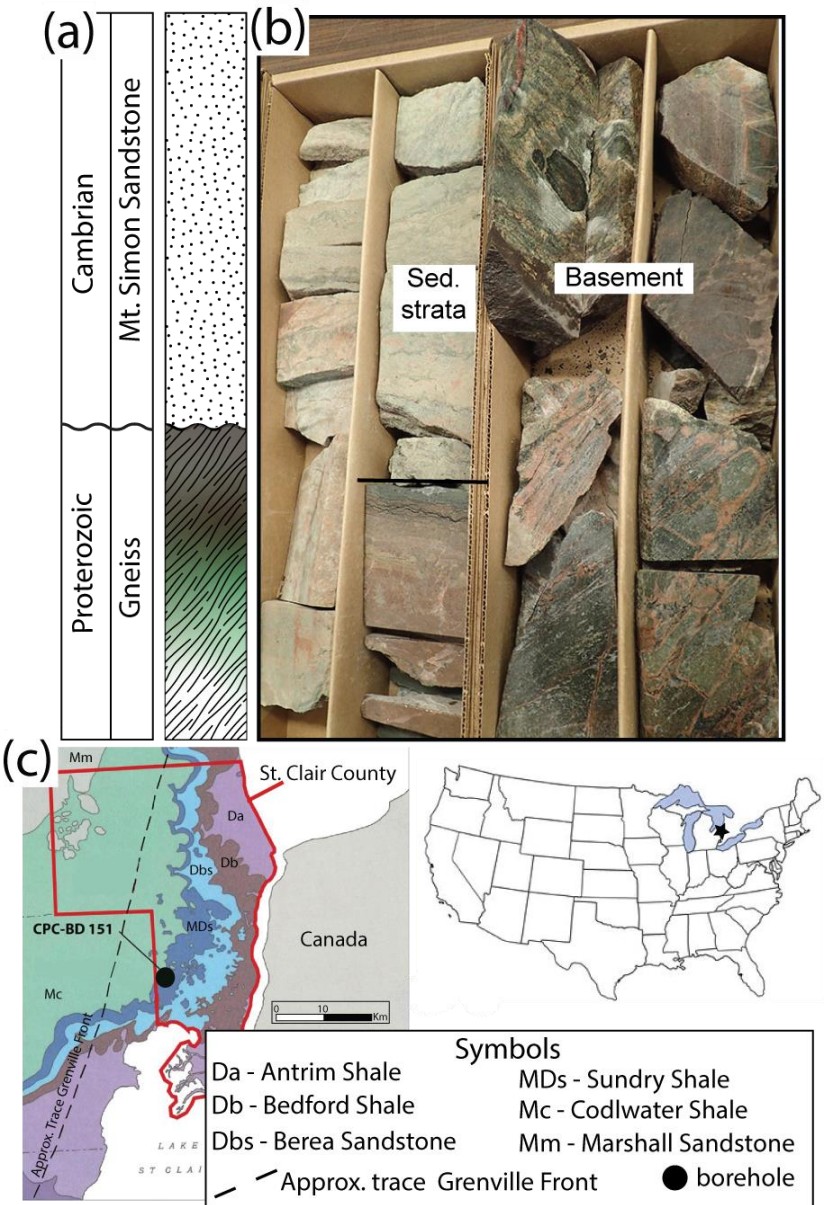

**Figure 5. A) Lithologic log of the CPC BD-139 core, Michigan from 1404-1412.1 meters measured depth. Five main lithologic units identified, including sandstone, dolomitized and undolomitized finely foliated gneiss, and dolomitized and undolomitized gneiss with sub-horizontal white veins. B) Photographs of the CPC BD-139 core. Core between ~1404.5-1405.5 meters. Contact between the Cambrian Mount Simon Sandstone (light tan) and the underlying Precambrian gneiss. The gneiss directly at the contact is fine-grained, tan, and dolomitized. This is underlain by green altered gneiss with sub-vertical pink fractures. This lithology grades into a dark grey gneiss with sub-horizontal white veins (core between 1411.5-1412.5 meters), which extends through the bottom of the logged section. C) Geologic map St. Clair County, Michigan, modified from Milstein, (1987).**

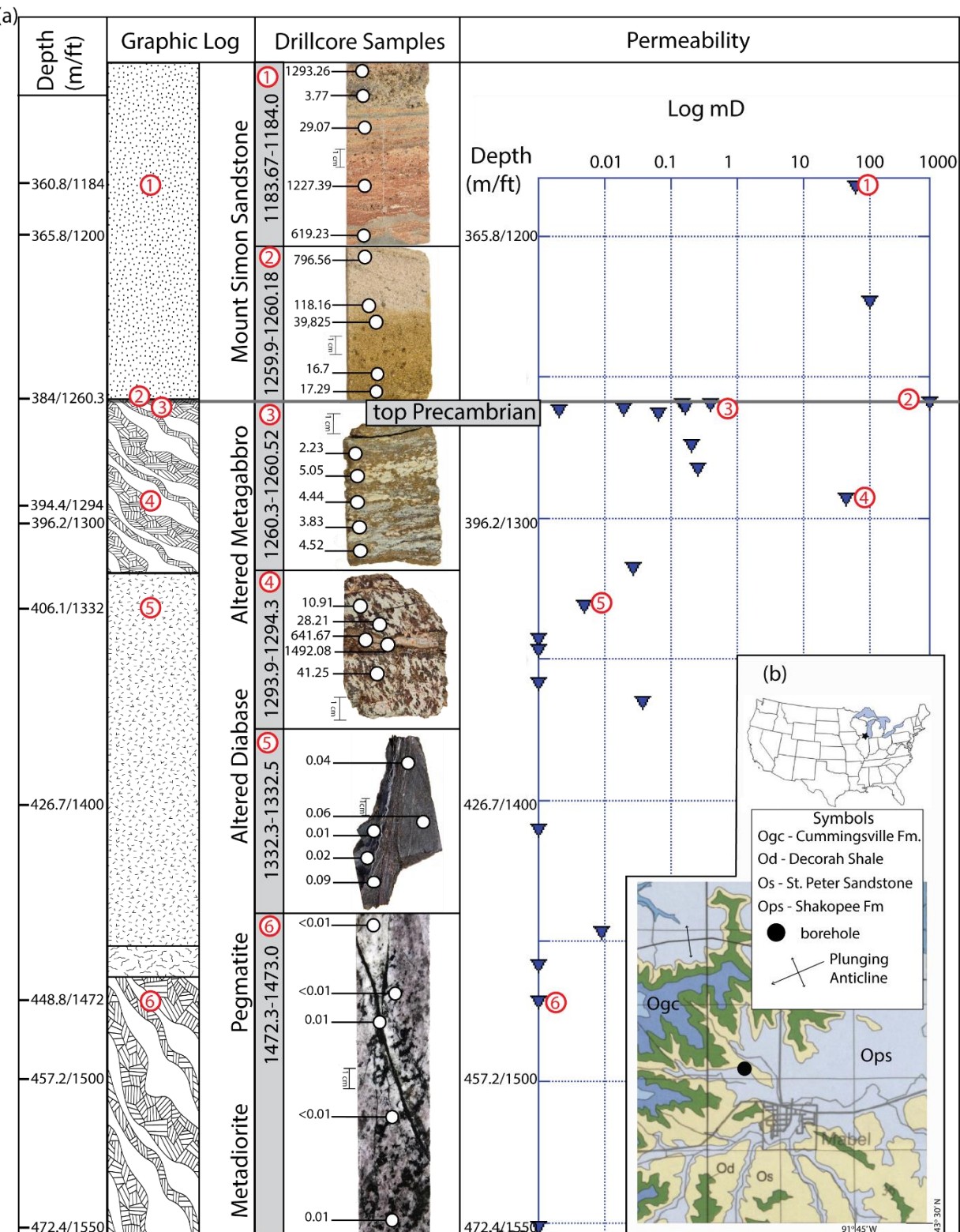

**Figure 6. A) BO-1 lithologic log with select representative core samples of each the major lithologic units. Above the nonconformity, the analogue reservoir or injection unit, the Cambrian Mount Simon Sandstone is porous with evidence for both dissolution and oxidation front. The crystalline basement rock consists of foliated, intensely altered and altered metagabbro with localized faulting,**
**variably altered and faulted diabase localized intrusions, pegmatite dikes, and at greater depths, relatively unaltered and less-deformed metadiorite. Gas permeability measurements were made on 25 core samples spanning the nonconformity interface. For each core sample tested, 5 spot measurements were made (locations shown by white circles). For relative comparison across the contact and within the various lithologic units, permeability data (millidarcy) is plotted using a log scale and the averaged values for each sample. B) Geologic map modified from Mossler, et al., 1995.**


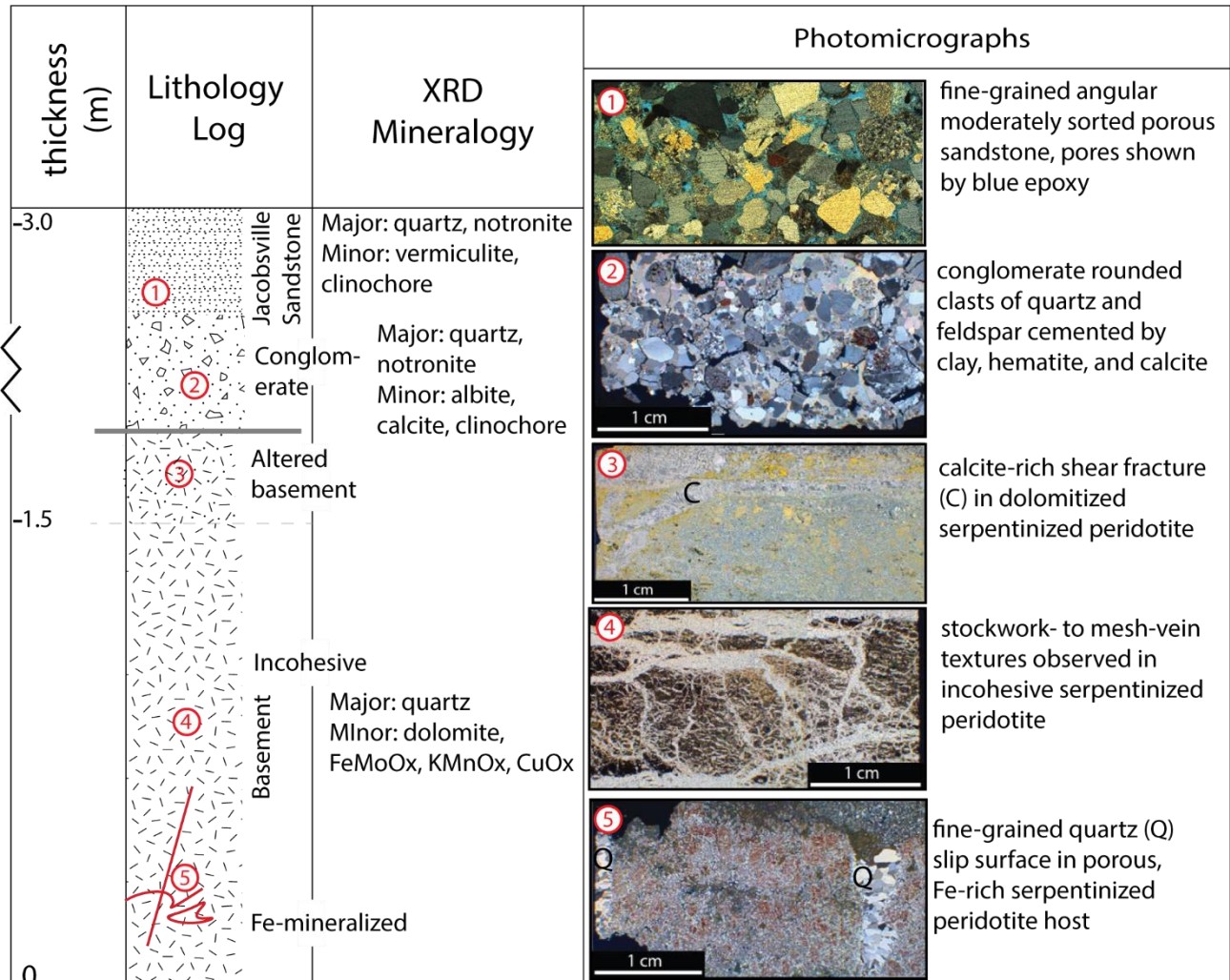

**Figure 7. Petrographic summary figure, photomicrographs and X-ray diffraction results of nonconformity units studied at Presque Isle, Michigan. 1) Jacobsville Sandstone arenite (100x, ppl) 2) Jacobsville Sandstone altered conglomerate (200x, ppl), 3) Basement**

calcite-rich slip surface in dolomitized, serpentinized peridotite (200x, xpl), 4) Basement serpentinized peridotite (100x, ppl), 5) Basement slip surface within in Fe-rich serpentinized peridotite (red) (100x, ppl).

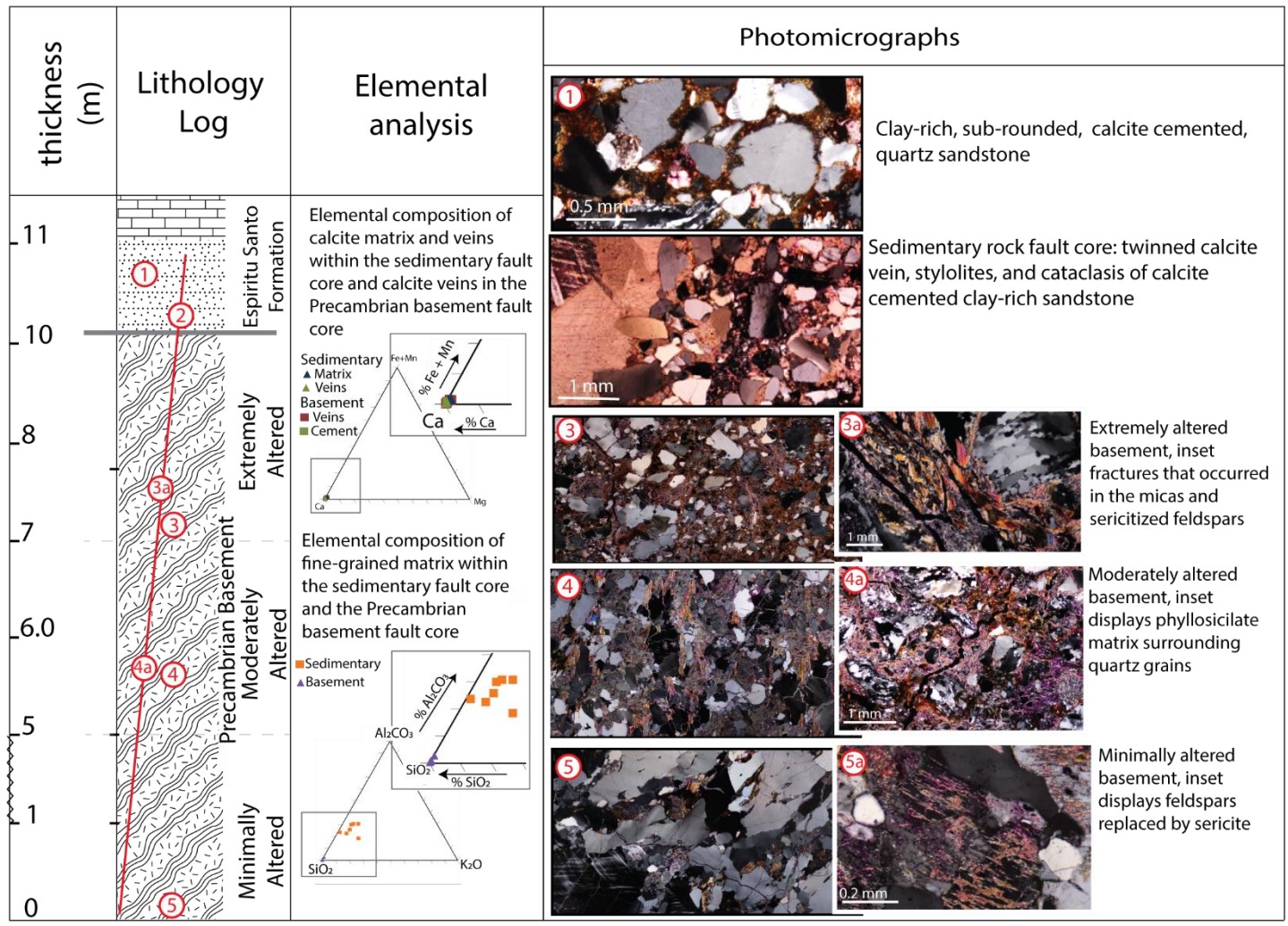

**Figure 8. Petrographic and elemental analysis summary of nonconformity units at Gallinas Canyon site. Thickness is measured in meters from base of outcrop section, red line represents fault. Elemental analysis shows similar calcite composition of the veins within the sedimentary sequence and the Precambrian basement faults. 1 & 2) The Espiritu Santo sandstones are clay rich calcite cemented quartz sandstones, in the fault core, 2) the sandstones are cut by twinned calcite veins and stylolitic textures and contact cataclastic; 3) adjacent to the nonconformity the granitic basement contain fractures in the micas and sericitized feldspars; 4) & 5) basement alteration decreases away from the nonconformity with phyllosilicate matrix surrounding quartz grains and sericitization of feldspars occurring 10 m from the nonconformity.**

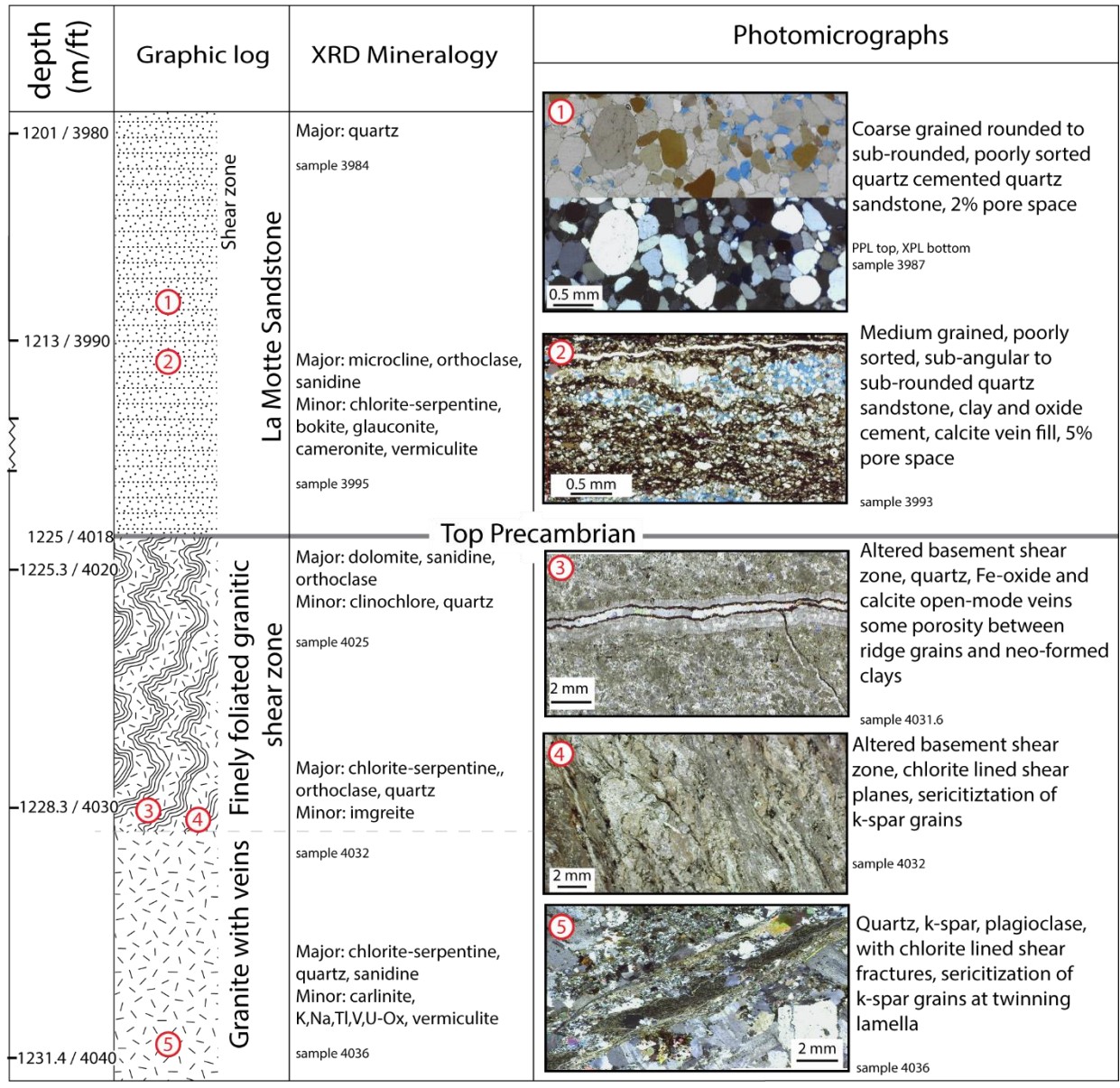

**Figure 9. Petrographic summary figure, photomicrographs and X-ray diffraction results of nonconformity units studied in the R.C. Taylor 1 core; 1) La Motte Formation sandstone, rounded to sub-rounded, poorly sorted quartz sandstones (100x, ppl & xpl); 2) lower La Motte Formation, opaque Fe-oxide cements and vein fill, porosity shown by blue epoxy, (100x ppl); 3) Top Precambrian crystalline altered basement shear zone. Syntaxial veins mineralized with quartz, reactivated and mineralized with Fe-oxide then sparry calcite. Some porosity between rigid grains and neo-formed clays (50x PPL); 4) Altered basement shear zone, chlorite lined shear planes, sericitization of feldspars along twining lamella (100x ppl); 5) Coarse crystalline sericitization of feldspars adjacent to twin lamellae (150x XPL).**

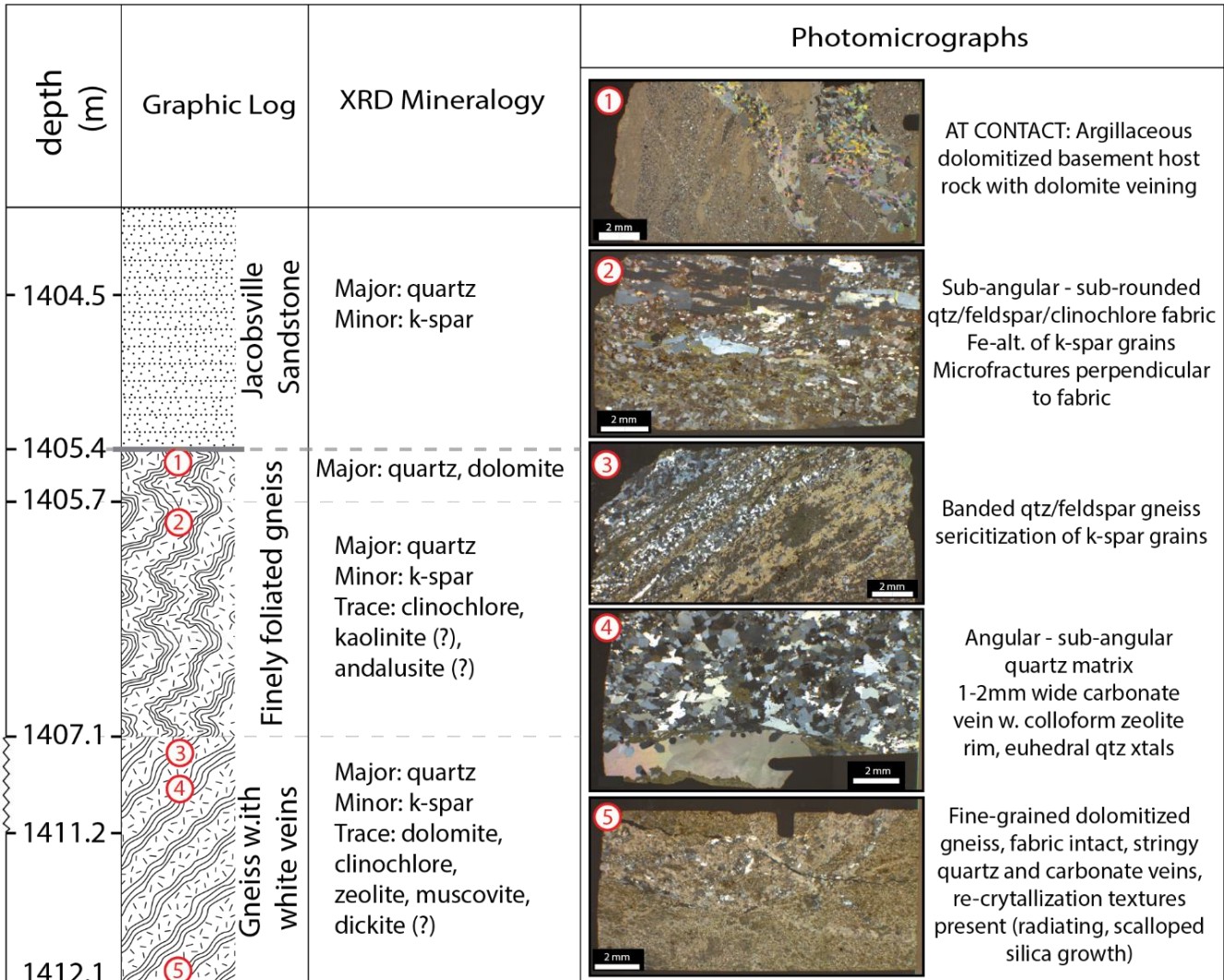

**Figure 10.** Petrographic summary figure, depth in meters measured depth, photomicrographs and X-ray diffraction results of nonconformity units studied in CPC BD-139 core. 1) Basement sample at the contact is an argillaceous dolomitized gneiss with dolomite veins (XPL); 2) Foliation defined by quartz-feldspar-clinochore fabric with iron alteration of potassium feldspar grains (XPL); 3) banded quartz-feldspar gneiss with common sericitization of potassium feldspar grains (XPL); 4) Carbonate vein with colloform zeolite rim and euhedral quartz crystals (XPL); 5) dolomitized gneiss with common quartz and carbonate veins (XPL).

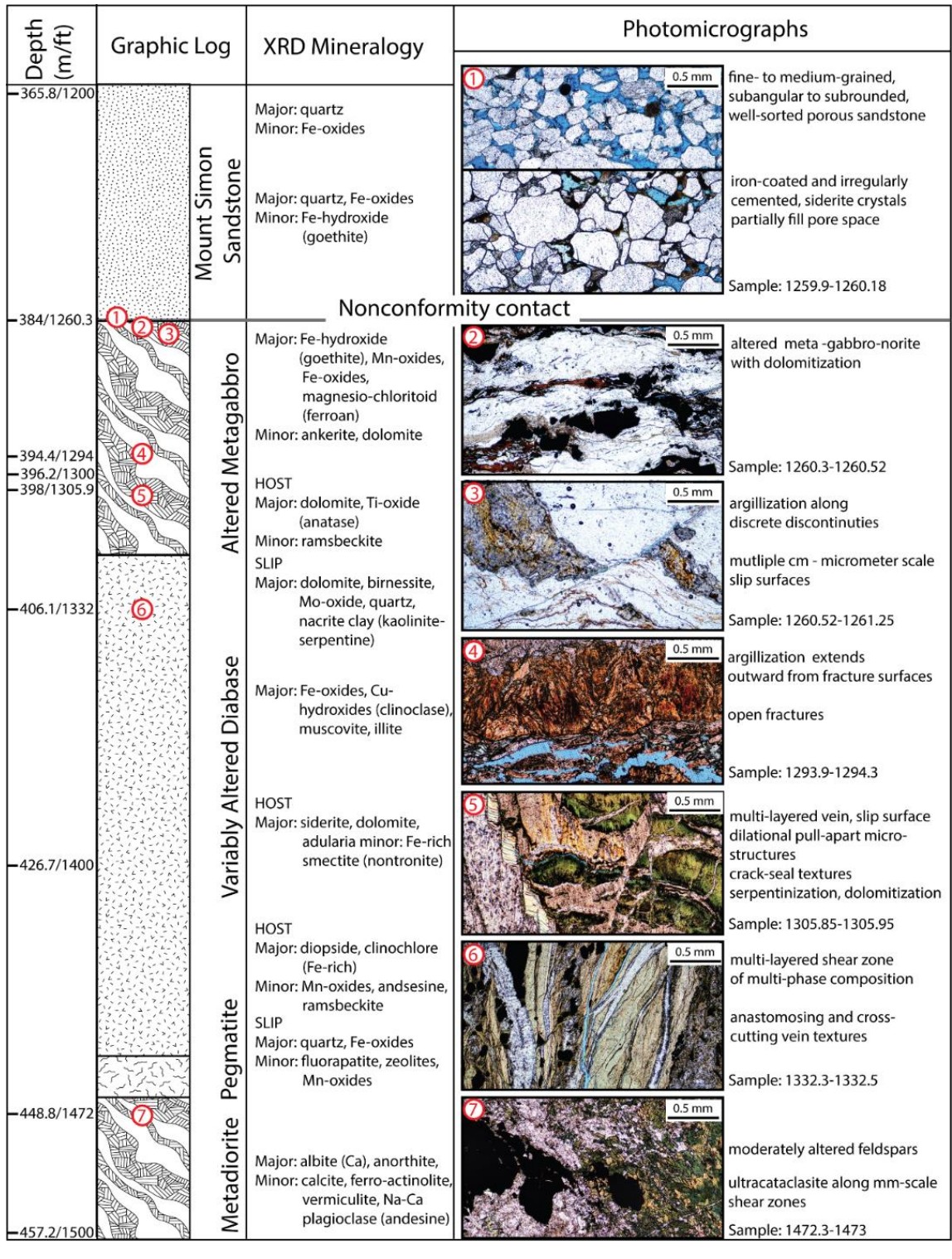

| Depth (m/ft) | Graphic Log | | XRD Mineralogy | Photomicrographs |
|---|---|---|---|---|

Depth (m/ft):
- 365.8/1200
- 384/1260.3
- 394.4/1294
- 396.2/1300
- 398/1305.9
- 406.1/1332
- 426.7/1400
- 448.8/1472
- 457.2/1500

**Mount Simon Sandstone**

Major: quartz
Minor: Fe-oxides

Major: quartz, Fe-oxides
Minor: Fe-hydroxide (goethite)

Nonconformity contact

**Altered Metagabbro**

Major: Fe-hydroxide (goethite), Mn-oxides, Fe-oxides, magnesio-chloritoid (ferroan)
Minor: ankerite, dolomite

HOST
Major: dolomite, Ti-oxide (anatase)
Minor: ramsbeckite
SLIP
Major: dolomite, birnessite, Mo-oxide, quartz, nacrite clay (kaolinite-serpentine)

**Variably Altered Diabase**

Major: Fe-oxides, Cu-hydroxides (clinoclase), muscovite, illite

HOST
Major: siderite, dolomite, adularia minor: Fe-rich smectite (nontronite)

HOST
Major: diopside, clinochlore (Fe-rich)
Minor: Mn-oxides, andsesine, ramsbeckite
SLIP
Major: quartz, Fe-oxides
Minor: fluorapatite, zeolites, Mn-oxides

**Pegmatite**

**Metadiorite**

Major: albite (Ca), anorthite,
Minor: calcite, ferro-actinolite, vermiculite, Na-Ca plagioclase (andesine)

Photomicrographs:

① 0.5 mm — fine- to medium-grained, subangular to subrounded, well-sorted porous sandstone

iron-coated and irregularly cemented, siderite crystals partially fill pore space

Sample: 1259.9-1260.18

② 0.5 mm — altered meta-gabbro-norite with dolomitization

Sample: 1260.3-1260.52

③ 0.5 mm — argillization along discrete discontinuties

mutliple cm - micrometer scale slip surfaces

Sample: 1260.52-1261.25

④ 0.5 mm — argillization extends outward from fracture surfaces

open fractures

Sample: 1293.9-1294.3

⑤ 0.5 mm — multi-layered vein, slip surface dilational pull-apart micro-structures
crack-seal textures
serpentinization, dolomitization

Sample: 1305.85-1305.95

⑥ 0.5 mm — multi-layered shear zone of multi-phase composition

anastomosing and cross-cutting vein textures

Sample: 1332.3-1332.5

⑦ 0.5 mm — moderately altered feldspars

ultracataclasite along mm-scale shear zones

Sample: 1472.3-1473

**Figure 11. X-ray Diffraction mineralogy and photomicrographs of BO-1 drill core samples showing representative compositions and textures across the non-conformity interface contact. Samples within centimetres of the contact (1-3) are strongly weathered, altered, and slightly metamorphosed gabbro-norite. Alteration and diagenesis assemblages include iron-oxides-hydroxides with chlorite, ankerite, and dolomite. Alteration extends for ~ 50 m into the basement. Sample 3 illustrates mm-scale offset across argillite layer. Note fracture permeability (blue-epoxy) parallel to slip surface. Fracture surfaces within Sample 4 at 34 m below the contact are several mm's of mixed chlorite-clay alteration and fine-scale permeability (blue-epoxy). Sample 5 at 1304.9 m shows multiple phases of fluid-rock interactions coupled with dilation, serpentinization, and dolomitization. Multi-layered clay-rich fault core gouge within Sample 6 at 1332 m or ~ 70 m below the contact. Note, open fractures within central portion of fault core gouge (blue-epoxy). At 1472 m or 272 m below the contact within the meta-grano-diorite unit, clay alteration is observed within feldspar grains at the micro-scale.**

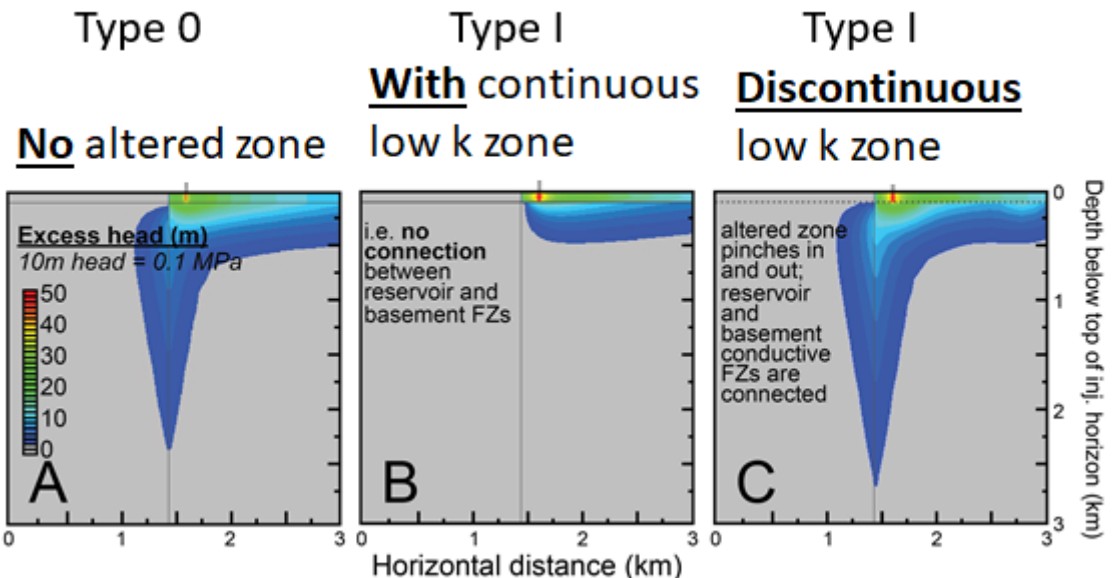

**Figure 12. Cross-sectional views of pore pressure envelope propagation resulting from injection into a reservoir underlain by a low permeability altered zone. Excess hydraulic heads after 4 years of constant-rate injection are presented for a Palaeozoic conduit-barrier fault scenario (A) Faulted Type 0 nonconformity, absent a low-permeability altered zone, (B) Type I nonconformity, with an altered zone present as a 20-m-thick confining layer (represented by two horizontal grey lines) that is continuous such that reservoir and basement fault zones do not connect, and (C) Type I nonconformity, with a discontinuous altered zone that pinches in and out in 20-m horizontal intervals (i.e. undulating) but where the reservoir and basement fault zones are fully connected. Results are zoomed in to the top 3 km × 3 km of the model domain. Vertical grey lines indicate location of the fault zone. Injection well**

location is indicated on top of each panel. Transition from grey to dark blue contour (and all subsequent contour lines) denotes a 2-m increase in hydraulic head (0.02 MPa). Adapted from Ortiz (2017; for details of the modelling approach see Ortiz et al., 2019).

**Figure 13. Proposed geologic schematics of the non-conformity contact region. A) Type 0 – sharp contact expected to prevent direct fluid pressure communication across the contact while promoting migration parallel to the contact distributing fluids laterally; B) Type I – phyllosilicate dominated zone above crystalline basement is expected to inhibit fracture propagation across the**
665 **nonconformity, prevent fluid migration due to permeability contrast and promote lateral migration; downward fluid migration can occur at a permeable fault zone; C) Type II – secondary mineralization dominated zone, lateral migration due to permeability contrast, mineralization due to fluid-rock interactions suggests that deep fluid circulation occurs even without enhanced permeability from fractures. All nonconformity types may be cut by structural discontinuities. Blue arrows indicate potential flow paths of injected fluids.**

**Table 1.**

| Nonconformity Type | Site Location | Summary Features |
|---|---|---|
| **Type O** | Hidden Beach Marquette, Michigan | Sharp, discrete non-conformity contact ± topographic variations between porous sedimentary sequences and non-porous crystalline basement |
| **Type I** | Gallinas Canyon, New Mexico  RC Taylor Core | Altered contact with phyllosilicate mineralization; faults within basement only and/or cross-cutting non-conformity contact into overlying sedimentary |

| | | rocks, open-mode veins in basement and overlying sedimentary rocks |
|---|---|---|
| **Type II** | Presque Isle, Marquette, Michigan<br>CPC BD-139 Core<br>BO-1 Core | Altered contact with non-phyllosilicate secondary mineralization, hydrothermal mineralization; faults within basement only and/or cross-cutting non-conformity contact into overlying sedimentary rocks. Open-mode and hybrid veins in basement and overlying sedimentary rocks |