# Peer review of "Geologic characterization of nonconformities using outcrop and core analogues: hydrologic implications for injection-induced seismicity"

_Solid Earth, 2020_

## Referee Comment (RC1) · Owen Callahan (Referee) · 21 Apr 2020

Review for Petrie et al., submitted, Geologic characterization of nonconformities using outcrop and whole-rock core analogues: hydrologic implications for injection induced seismicity. Solid Earth

Reviewed by Owen Callahan, April 20, 2020

General Comments: The authors present important and timely work about rock properties at the basement-basin interface and discuss potential implications for induced

seismicity. However, the manuscript is missing a higher level synthesis needed to make the observations useful in a more practical, applied, or global sense. I encourage the authors to consider restructuring the paper so that their important observations are highlighted and lead to more transparent interpretations, with more explicit discussion of the impact of local geologic history or other possible controls on the properties of the non-conformity. I think these changes would make the paper much more impactful.

Specific Comments: 1. The manuscript is missing a higher level synthesis about what (predictable) factors control the various types of interfaces. In this regard, the manuscript feels more like a list of observations, without a real test of the hypothesis or further prediction. For instance, what is the impact of specific lithologies on fluid flow properties at the interface? Are fault damage zones different in the different basement lithologies? Is the weathering (paleosol) different depending on lithology, exhumation, or burial history? This important step would help identify other regions more or less prone to permeable basement interfaces based on knowable basement properties.

2. Paper organization: I think the paper would benefit from more informative headings. Methods could be more detailed. If modeling were its own section, not part of the discussion, then the model results could be used to support specific points in Discussion (referring to Fig 12). The Introduction is a bit jumbled. The manuscript is missing a section on local geology descriptions, which could be very informative when trying to generalize about geologic controls on the basement-basin interface. Instead these observations are lumped into results; this has the effect of making site selection seem opportunistic rather than designed to test specific differences in geology or geologic history, because the differences between sites or reason for picking them is not clear before the results are reviewed.

Line Item and Technical Comments: 20: Capitalize "Great Unconformity" 24: Perhaps just name the types here, rather than write around them? 26: Which one, and why? What contributes to allowing or inhibiting? 43: Missing second parenthesis. 45: This paragraph outlines the results from this study, but, due to its position in the introduction, makes it sound like a prior observation or known phenomena. I think it either needs more citations, or to be moved later into the intro as part of what you are describing as your work and the results of this study. 60: Whole-rock core here (and in the title) seems a bit redundant. Is this to differentiate between core and cuttings? 73: . . .observed/described rock AND fracture and fault features. Without introducing observations of fault/fractures earlier, the descriptions at each site seem to come out of the blue or to be irrelevant to the main hypothesis, even though faults and fractures in the basement are clearly very important. 77-84: This is a broad site description, not really methods. I would suggest including a more generic section on Geologic Setting (which could include much of what you describe in results), expanding the methods, and streamlining results so the reader can more easily map to your synthesis diagram in Fig. 12. 86-87: Analytical methods could be more detailed, or reference details in archived dataset. The specificity of the mineral identification from XRD is impressive, so it would be nice to know what kind of equipment, scan times, and analysis software was used. 125: Regarding granular flow, could use citation. 178: "altered and altered" 180: missing a period after (Anderson, 2012) 180-183: This statement would lead naturally to a broader synthesis discussion point about the nature of the nonconformity in mafic, rift-related portions of the midcontinent. 183: Looks like a missing space after "1995)."? 192: The paragraph break here is confusing for me, as the topic sentence is about the Mt. Simon, but the next few sentences are again referring to the altered upper portion of the basement, but that is not entirely clear until the reference to "50 m zone. . ." 199: Either new sentence or semi-colon at "contact, locally. . ."? 223: The observation that the altered shear zone can be fractured/reactivated would be another point to include in a broader synthesis: zones of prior deformation are more likely to be zones of subsequent deformation 229: Another good point to fold into a broader synthesis: phyllosilicates at the contact may inhibit cross-nonconformity fractures and flow, but maybe need to discuss their origin. 225-242: I find this section somewhat confusing. Doesn't the alteration and mineralization suggest fairly extensive fluid-rock interaction? I think the part that is missing is the point that prior fluid rock interaction

and alteration has resulted in low permeability now, so perhaps clearing up the temporal aspects? However, mineralization (presumably strengthening) and alteration to phyllosilicates (presumably weakening) are both called upon to act as hydrologic and mechanical barriers, and as written it is hard to understand why. Consider, for instance, the impact of more brittle layers in fault systems (e.g. Schöpfer, M. P. J., Childs, C., & Walsh, J. J. (2006). Localisation of normal faults in multilayer sequences. Journal of Structural Geology, 28(5), 816-833. 10.1016/j.jsg.2006.02.003) 259: "Our collective field and core observations document the occurrence of significant lateral variations in altered or mineralized zones that are associated with a relatively wide range of permeability values, and that alteration coupled with abundant structural discontinuities can result in relatively higher permeability that extends for 10's of m's both laterally and vertically into the crystalline basement rock below the nonconformity." Yes, but what controls the variations, and how might someone know from the surface, prior to siting an injection well, if basement faults in a region are more or less likely to be reactivated due to hydrologic properties at the non-conformity? This is the type of synthesis I would really like to see spelled out more explicitly, even if speculative, and there are sections where you already briefly bring up points that could feed into this broader synthesis (see above). 264: Modelling work could be a new method and result, then folded into discussion and used to support your summary diagram in Fig. 12, rather than added to the end. As it is, I find it hard to tell if this is new work, or prior work from J. Ortiz. 266-267: Note different style "x" shown for "times" in permeability values. Also, is the relative kz/kx and absolute kz shown meant to be for the fault in basement or in the aquifer, because wouldn't at least one of those values need to be different for the two portions of the faults? 271: Ok, so in this model the low permeability zone formed after the fault? Or was it a weathered horizon that was incorporated into the fault? I wonder because that helps me think about the environment of formation of this zone, which could be very widespread at non-conformities (e.g. soil profiles like Walter et al. (2018) Petrophysical and mineralogical evolution of weathered crystalline basement in western Uganda: Implications for fluid transfer and storage. AAPG Bulletin). 283: I would like to see these

geologic conditions spelled out more explicitly rather than leaving it to the reader to infer them. 284-286: In what scenario would they do both, either? This sentence could be more specific. 288: Do you mean Type 0 and Type 2? I thought Type I resulted in reduced fluid communication into the fault zone (line 274-275)? 295: Shameless plug about impact of fluid chemistry on deformation in mineralized fault rocks, but there are many others too: Callahan, O. A., Eichhubl, P., Olson, J. E., & Davatzes, N. C. (2020). Experimental investigation of chemically aided fracture growth in silicified fault rocks. Geothermics, 83. https://www.doi.org/10.1016/j.geothermics.2019.101724 359: Looks like a citation manager software glitch. 363, 364: Incomplete references? No pages or publisher? Might just be a reference style thing. 385/Figure 1. In legend, using a grey gradient box for 'craton' would be a bit clearer than the current black line, although this may be a reproduction issue. 390/Figure 2. Missing a description of inset "A". Not clear what diagonal lines are in B, Fault? Dike? Caption could be more informative, for instance noting evidence for fluid-rock interactions in "C". Some shorthand in captions is confusing, such as "min. congl". Mineralized? Minimal? 395/Figure 3. References to insets change from 1) , 2) to B), C), . . . "Colloform mineralization" image ("C") is either missing or does not chow colloform habit very clearly. What are red lines near 4 and 5 on the "Lithology" log? Why does the thickness start with 0 below weathered basement and not at the non-conformity? 397/Figure 4. Could use more descriptive text, for instance, insets A, B, are not discussed. Red arrows in B, C are not described. Fault in C would be easier to see if white or other light color. B would benefit from a scale. 400/Figure 5. Choice of height scale at 1.2 m is a bit odd, and "Thickness" may be the wrong word to use here. Maybe depth relative to non-conformity and start with 0 there? Is Espirutu $\sim$10 m thick, or 10.9-9.6 m thick? Because it shows very little, the "Elemental Analysis" column is a bit frustrating. Why not show XRD results as in Figures 3 and 7? Or better yet in all figures with similar columns show alteration/mineralization reactions and products, which could reflect either XRD or elemental work (e.g. + calcite, + albite, -quartz. . .) if you have a mixed bag of analyses. Note typo in 4a "phyllosicilate". In caption: Espirutu or Espiritu? 408/Figure 6: This caption has a lot of passive voice.

Unclear to me if the nonconformity is cut by "throughgoing veins" (check typo there) or the veins cut the shear zone, but those scenarios have pretty different implications. 414/Figure 7: Perhaps worth noting whether units are measured depth, relative to sea level, true vertical depth, etc. for clarity. 425/Figure 8. Typo: "Granitiod" in A. "Pink-coated" is not particularly helpful; perhaps ID the mineral, even if speculative, or just call them partially-mineralized, sealed, or stained fractures, etc. whatever the case may be. 430/Figure 9. Same comment about measured depth for clarity. 435/Figure 10. Cool plot. The "Depth" axis on the permeability column is perhaps redundant. 444/Figure 11. Photomicrograph 2. What does "intensely weathered ∼60 m" mean? Consider argillic alteration or just argillization. Argillite is a rock type, not the product of argillic alteration. "Iron" capitalized in caption. 446/Figure 12.... "Phyllosilicate" (in figure) vs "weathering" in caption. What is the rationale for the circular flow path? You made it this far! Good luck!

---

## Editor Comment (EC1) · Roger Soliva (Editor) · 21 Apr 2020

Dear Authors,

We have received the first reviewer comments and I wait now for the second review to get an overview of the paper and provide general recommendations.

Best regards, Roger Soliva
* * *

---

## Referee Comment (RC3) · Anonymous Referee #2 · 5 May 2020

The paper documents nonconformities that are associated with Precambrian crystalline rocks overlain by porous sedimentary rocks. Results are based on observations from two outcrops and three cores. Data on the mineralogic and structural heterogeneities are provided to discuss the impact of nonconformities on fluid flow. I think that the paper is valuable for studies related to nonconformities and is topical in the context of induced seismicity due to fluid injection. My major points are detailed below, and I recommend acceptance of this manuscript after a major revision.

Specific comments

1. Illustrations of the fractures. The figures largely focus on the petrographic features and I found that the fracturing mentioned in several places is not fully illustrated. This includes, for example, (1) the near-vertical to bedding-parallel bleached fractures (l.100), (2) basement-hosted slip surfaces (l.101), cm- to m's - displacement faults (l.115), slip surfaces with oblique to dip-slip slickenlines mm's to cm's thick (l.189)..... I think the manuscript will be improved by illustrating these features.

2. Lateral variability. Nonconformity zones likely have properties that are spatially heterogeneous. This is indicated l.49-52: "Due to weathering, deformation, diagenesis and fluid-rock interactions, the nonconformity zone may be hydraulically heterogeneous at the mm to 10's m scales". The paper addresses the vertical variability, but it does not fully address the lateral variability, which is critical for fluid flow and can give insights into how core data can be extrapolated to large-scale application. The studied outcrops allow analysing this lateral variability. For example, rocks studied in section 2.2.2 are observed along a 4-km long section in Gallinas Canyon. I will recommend the authors to further this point.

3. Geological settings. The description of the geological setting is very sparse. I recommend the authors to provide more information for each studied area. For example, the types and ages of the fractures and some general descriptions on the tectonic setting. Maybe providing the locations of the studied areas on geological maps could be useful.

4. Analogues. Two outcrops are examined: (1) a nonconformity between late Proterozoic Jacobsville Sandstone and early Proterozoic altered peridotite outcropping in Michigan and (2) the contact between Devonian to Mississippian carbonate and clastic rocks of the Espiritu Santo Formation deposited on the Proterozoic quartzofeldspathic and amphibolitic gneiss. Also, three cores are examined: (1) a core recovered from the Cambrian Lamotte Formation sandstone and sheared Proterozoic granitoids in south-central Nebraska, (2) from the Cambrian Mt. Simon Sandstone and Precambrian altered granitoid gneiss of the Grenville Front Tectonic Zone and (3) from a section of

rocks of lower Cambrian Mt Simon Sandstone overlying a Precambrian layered intrusive complex. I think it is interesting to analyse these very different areas because it provides an idea about the diversity of the nonconformity. However, the negative point is that it is not clear whether the studied areas are analogues to fluid injection sites or not. For example, in the introduction, the authors mentioned the mid-continent United States and the works by Murray (2015). This work concerns Oklahoma's underground injections, where target rocks are mostly carbonates from the Arbuckle sedimentary strata above a crystalline basement. My knowledge of the regional geology of the studied areas is limited, but I think it will be worthy to justify the choice of the studied areas and how these areas are relevant for fluid injection operations. I think the last conclusion point: "the contact . . .. should instead be evaluated on a site by site basis prior to injection of large fluid volumes" is critical in this regard.

5. Quantitative insights. The authors briefly describe permeability values measured for one core in section 2.2.4. This is important data and the results could be described further. Besides, I think it could be interesting to provide similar information from the other studied areas. Also, there is a mention of fractures with density decreasing with depth (l.189). I think this could be described quantitatively as well. For example, by providing a fracture density log. More generally I think providing further quantitative analysis will be welcome and will make this work more valuable.

6. Numerical modelling. I think the last paragraph of the discussion on numerical modelling will be better in the result section. Also, I think that the authors should provide more information about their methodology, the origin of the permeability values, the choice of the studied geometry and the boundary conditions.

Minor points

1. Mt. Simon or Mount Simon, both are used in the manuscript.

2. l.204 "not the result of alteration due to weathering alone". I think this should be discussed further.

3. The figure captions are often incomplete.

4. Fig.2: What is A in the figure. There is no scale in B and C.

5. Fig. 3. There are no B, C and D in Fig. 3. I think the authors mean 4, 5 and 6. Although there is no 6 in the figure?

6. Fig. 4: What is A, B and C. There is no scale in B.

7. Fig. 6: B is not indicated in the caption.

I hope this will help to improve the manuscript.

---

## Referee Comment (RC4) · Anonymous Referee #3 · 5 May 2020

The topic of the geological attributes of rocks near cover-basement contacts is of topical interest owing to issues related to fluid injection and induced seismicity. The paper is well within the scope of the journal. It's a well presented and clearly illustrated paper.

The Introduction should be improved by adding a specific statement of a claim or claims for the paper. We know from the Introduction that the attributes of the basement-cover interface zone is important and we have an outline or agenda for what nonconformities were examined and how. But there is no statement of the 'here we show that' variety to motivate the reader to read through the details. Such a statement should be

added. There are several places in the text where vague or ambiguous usages could be improved.

The Discussion presents some inferences about fluid circulation and the interpretations of structures and mineral deposits. As it stands some of this text seems speculative. The arguments should at least be bolstered by pointing to some of the extant structural diagenesis literature.

Where the text describes 'fractures' and fracture mineralization, the descriptions could be more complete (and meaningful). More information could be provided on whether the fractures are 'opening mode' or faults. The use of the term 'vein' is unhelpful, particularly with respect to structures in the cover above the nonconformities. Mineral fill in fractures is common throughout sedimentary sequences (e.g. Laubach et al. 2019, Reviews of Geophysics) and such mineral deposits could provide evidence of the post depositional structural and fluid history of these zones. So a more meaningful description of these features could be useful. Note also that there are a number of published studies of fracture systems in basal Cambrian and in Ordovician sandstones of the midcontinent and other Laurentia cover rocks, and the fracture sets have a range of ages and origins. Some statement as to how representative these outcrops are of the midcontinent nonconformity zones would be helpful.

I think I follow what you are saying here about the definition of the 'nonconformity zone', but perhaps the definition could use sharpening. Are you talking about some volume of rock near the nonconformity that is somehow altered from what it would be if the same rock was not near the nonconformity? Do you only mean rocks in the basement or could this include rocks above the nonconformity? Can you try to make the definition more explicit?

Where you mention 'the nonconformity' it might help reader if you remind them here that you mean 'the nonconformity in the US midcontinent region'.

The Introduction would be improved by adding an explicit claim here that could start with the statement 'here we show that. . .' Motivate the reader rather than just providing a list of what you did.

But are these overlying rocks mostly quartz-rich sandstones? Isn't the basal Cambrian sandstone pretty common? I see that you outline the geology you looked at in section 2.1. Do you discuss how representative these might be?

Where in the Introduction do you alert the reader that you present modeling?

'detailed' is vague; can you replace this statement with a scale (or range of scales)? Or just omit, since the resolution level is implied by the instruments you used.

Is there a reason for the order that you describe the localities? Same question for the listing in section 2.1. A representative selection?

How low is the porosity?

if the fractures are bedding parallel as you say, it would be hard for them to extend into basement. Or do you mean the reduction spots are not in basement?

Are these slip surfaces in basement subparallel to the bedding parallel 'fractures' in the cover. Are the cover fractures faults?

By 'span the contact' do you mean the faults extend into the cover?

Something is awkward in the phrasing here.

Are you saying fault rock is only found in faults? Clarify text.

Quartz lined and quartz-filled fractures are common in quartzose sandstones even distant from nonconformities. The mineral deposits may not necessarily represent mineralization 'events' since the fractures themselves are reactive surfaces (e.g. Lander and Laubach 2015, GSA Bulletin).

and preceding text. What kind of 'fractures'; opening mode, or faults? Are there crosscutting relations here that provide evidence for the relative timing of these structures? Are you implying that the shear zone in the basement is somehow related to the fractures in the cover? (Wouldn't that be surprising?)

Is this the porosity range at the site you sampled? It seems a stretch to say that this is the range for the Mt Simon generally, since porosity ought to reflect thermal exposure/burial history and that could vary regionally. Clarify.

space

'multi-layered veins and/or fracture mineralization'; are these different things?

'porous'; but can you specify how porous?

'structural discontinuities' seems vague.

Is the thickness of the nonconformity zone specified at the outset of each description above? And how did you decide where the boundaries of the zones are?

What is the opposite of 'in situ' mineral growth?

Maybe put in a table? And refer to in description.

The first paragraph of the Discussion seems vague and disorganized. Are these structures in the nonconformity zone' or in the basement or the cover? Are these only 'small faults' or are some of the fractures opening mode?

The 'non fractured'; do you mean that these zones lack fractures in general, or that in areas where fractures happen to be absent, the host rock attributes might have these effects?

'we note that. . .'; what is the basis for this inference? That there are porous rocks above the basement rocks?

219-220 I don't see how it follows that the 'vein mineralogy' provides evidence for cross unconformity flow. Are you talking about mineral filled fractures in the basement or in the cover? Note that from mineral composition alone it can be challenging to find evidence for fluid flow (see for example, Denny et al.2020 GSA Bulletin). Maybe this point needs more development or the conclusion should be presented in a more nuanced way.

In the older rock mechanics literature there are examples of fracture systems in basement associated in typical midcontinent crystalline rocks that extend to depths of hundreds of meters and then abruptly stop; so zones of penetration of alteration could be much more than 5 m (and might be heterogeneous, if linked to deep seated fractures). Se references by Aubertin.

'that impacts' or 'that would be expected to impact'?

But are these the 'bed parallel' fractures?

What do you mean by 'deep circulation'? The basement rocks are not all that far from porous sedimentary rocks, which likely contain fluids.

Where did you mention what the mechanical properties of these rocks is? Did you measure them, or is that an inference from the rock types? An example of mechanical properties inhibiting fracture in the setting you are concerned with is in Ellis et al. 2012, J. Geol. Soc. London.

do you mean 'faults'?

259-264 Is this your claim?

Is this modeling work prefigured in the Introduction?

How representative are these various types you identify?

'Laubach' is the correct spelling.
* * *

---

## Editor Comment (EC2) · Roger Soliva (Editor) · 5 May 2020

Dear authors,

We have now the feedback from the reviwers. They both suggest major revision and I give you here a synthesis of the main points to consider to guide and facilitate your revision work and your answers.

1 - Introduction: As mentioned by R3, add a specific statement of a claim(s) of the paper, explicitly mentioning what you show exactly in this paper that help to improve

the knowledge of this topic. Also justify at the end of the introduction why you chose these examples, and especially why they are relevant to fluid injection operations as mentioned by R2.

2 - Reorganize the paper and especially add a section on local geology descriptions for each sites, which is mentionned by both reviewers. Also consider if it would be better to place the numerical modelling part in the result section as suggested by R1.

3 - Better synthesize, hierarchize and organize your observations (avoid listings) as suggested by R1. Highlight the most important observations that are used in the interpretations and better consider them in an applied or global sense in the discussion section.

4 - Better illustrate the fractures, veins and faults with photographs and add more precise descriptions (quantitative if possible) with relevant terminology as mentionned by R2 and R3.

5 - Give more information about the lateral variability of the non-conformities as suggested by R2.

6 - Avoid speculation about fluid circulation and diagenesis in the discussion section as mentionned by R3. Intergrate the relevant literature to give support to these interpretations.

Each reviewer also did a number of specific comments referring to line numbering that merit consideration in your reply.

Thank you in adavance for thoroughly considering and replying to each of these points and to provide an improved version of the manuscript.

Best regards, Roger Soliva
* * *

---

## Author Comment (AC1) · 15 Jun 2020

Line Item and Technical Comments: 20: Capitalize "Great Unconformity" – change made

24: Perhaps just name the types here, rather than write around them? - change made

26: Which one, and why? What contributes to allowing or inhibiting? – change made

43: Missing second parenthesis. – change made

45: This paragraph outlines the results from this study, but, due to its position in the introduction, makes it sound like a prior observation or known phenomena. I think it either needs more citations, or to be moved later into the intro as part of what you are describing as your work and the results of this study. – re-organized as per R1,R2,R3 comments (55)

60: Whole-rock core here (and in the title) seems a bit redundant. Is this to differentiate between core and cuttings? – change made

73: : : :observed/described rock AND fracture and fault features. Without introducing observations of fault/fractures earlier, the descriptions at each site seem to come out of the blue or to be irrelevant to the main hypothesis, even though faults and fractures in the basement are clearly very important. – We have added details earlier in manuscript

77-84: This is a broad site description, not really methods. I would suggest including a more generic section on Geologic Setting (which could include much of what you describe in results), expanding the methods, and streamlining results so the reader can more easily map to your synthesis diagram in Fig. 12.

We have included a brief section (section 2) that describes the geologic setting of each site.

86-87: Analytical methods could be more detailed, or reference details in archived dataset. The specificity of the mineral identification from XRD is impressive, so it would be nice to know what kind of equipment, scan times, and analysis software was used. – Details included.

125: Regarding granular flow, could use citation. – change made

178: "altered and altered" – change made

180: missing a period after (Anderson, 2012) – change made

180-183: This statement would lead naturally to a broader synthesis discussion point

about the nature of the nonconformity in mafic, rift-related portions of the midcontinent. This statement has been incorporated into the geologic setting and used in synthesis within the discussion.

183: Looks like a missing space after "1995).""? – change made

192: The paragraph break here is confusing for me, as the topic sentence is about the Mt. Simon, but the next few sentences are again referring to the altered upper portion of the basement, but that is not entirely clear until the reference to "50 m zone: : :" – added clarification

199: Either new sentence or semi-colon at "contact, locally: : :"? – change made

223: The observation that the altered shear zone can be fractured/reactivated would be another point to include in a broader synthesis: zones of prior deformation are more likely to be zones of subsequent deformation –

Noted. We have revised discussion to synthesize observations more clearly.

229: Another good point to fold into a broader synthesis: phyllosilicates at the contact may inhibit cross-nonconformity fractures and flow, but maybe need to discuss their origin.

Noted. We have revised discussion to synthesize observations more clearly in discussion

225-242: I find this section somewhat confusing. Doesn't the alteration and mineralization suggest fairly extensive fluid-rock interaction? I think the part that is missing is the point that prior fluid rock interaction and alteration has resulted in low permeability now, so perhaps clearing up the temporal aspects? However, mineralization (presumably strengthening) and alteration to phyllosilicates (presumably weakening) are both called upon to act as hydrologic and mechanical barriers, and as written it is hard to understand why. Consider, for instance, the impact of more brittle layers in fault systems (e.g. Schöpfer, M. P. J., Childs, C., & Walsh, J. J. (2006). Localisation of
normal faults in multilayer sequences. Journal of Structural Geology, 28(5), 816-833. 10.1016/j.jsg.2006.02.003)

Schopfer and other workers (Ferrill et al , Petrie et al, Larsen, Sibson, and others) show the change in failure mode across boundaries, our observation is that under the conditions in which the observed open-mode fractures formed they did not penetrate the nonconformity, reducing a potential future fluid flow pathway, and we expect the difference in relative permeability between the altered boundary and overlying sed. protolith injected fluids would move along the nonconformity. We have reworded this section to clarify.

259: "Our collective field and core observations document the occurrence of significant lateral variations in altered or mineralized zones that are associated with a relatively wide range of permeability values, and that alteration coupled with abundant structural discontinuities can result in relatively higher permeability that extends for 10's of m's both laterally and vertically into the crystalline basement rock below the nonconformity." Yes, but what controls the variations, and how might someone know from the surface, prior to siting an injection well, if basement faults in a region are more or less likely to be reactivated due to hydrologic properties at the non-conformity? This is the type of synthesis I would really like to see spelled out more explicitly, even if speculative, and there are sections where you already briefly bring up points that could feed into this broader synthesis (see above). We have provided clarification and edited the text; other reviewers have requested that we not speculate.

264: Modelling work could be a new method and result, then folded into discussion and used to support your summary diagram in Fig. 12, rather than added to the end. As it is, I find it hard to tell if this is new work, or prior work from J. Ortiz

We have reorganized the inclusion of the modeling work, models were built based on observations presented in this manuscript and have been presented in part by previous work by Ortiz. The models serve to test the impact changing characteristics of the

nonconformity have on fluid pressure migration.

267: Note different style "x" shown for "times" in permeability values. Also, is the relative z/kx and absolute kz shown meant to be for the fault in basement or in the aquifer, because wouldn't at least one of those values need to be different for the two portions of the faults?

We fixed the x; and permeability values are different (kx = kz = $3 \times 10^{-17}$ m2) in crystalline basement rock vs. conduit-barrier fault (kz/kx=105; kz = $3 \times 10^{-10}$ m2

271: Ok, so in this model the low permeability zone formed after the fault? Or was it a weathered horizon that was incorporated into the fault? I wonder because that helps me think about the environment of formation of this zone, which could be very widespread at non-conformities (e.g. soil profiles like Walter et al. (2018) Petrophysical and mineralogical evolution of weathered crystalline basement in western Uganda: Implications for fluid transfer and storage. AAPG Bulletin).

The fault cuts low permeability zones creating a potential permeability pathway.

283: I would like to see these geologic conditions spelled out more explicitly rather than leaving it to the reader to infer them. - revised

284-286: In what scenario would they do both, either? This sentence could be more specific. - revised

288: Do you mean Type 0 and Type 2? I thought Type I resulted in reduced fluid communication into the fault zone (line 274-275)?

The faults cutting Type 1 appear to be potential permeability pathways.

295: Shameless plug about impact of fluid chemistry on deformation in mineralized fault rocks, but there are many others too: Callahan, O. A., Eichhubl, P., Olson, J. E., & Davatzes, N. C. (2020). Experimental investigation of chemically aided fracture growth in silicified fault rocks. Geothermics, 83.

https://www.doi.org/10.1016/j.geothermics.2019.101724 359: Looks like a citation manager software glitch. - included

363, 364: Incomplete references? No pages or publisher? Might just be a reference style thing. 385/Figure 1. In legend, using a grey gradient box for 'craton' would be a bit clearer than the current black line, although this may be a reproduction issue.

We have modified the Basemap Figure (Figure 1) to support the newly added geologic setting details.

390/Figure 2. Missing a description of inset "A". Not clear what diagonal lines are in B, Fault? Dike? Caption could be more informative, for instance noting evidence for fluid-rock interactions in "C". Some shorthand in captions is confusing, such as "min. congl". Mineralized? Minimal? - change made

395/Figure 3. References to insets change from 1) , 2) to B), C), : : : "Colloform mineralization" image ("C") is either missing or does not chow colloform habit very clearly. What are red lines near 4 and 5 on the "Lithology" log? Why does the thickness start with 0 below weathered basement and not at the non-conformity?

We modified scale, added text to explain Fe mineralization.

397/Figure 4. Could use more descriptive text, for instance, insets A, B, are not discussed. Red arrows in B, C are not described. Fault in C would be easier to see if white or other light color. B would benefit from a scale. – change made

400/Figure 5. Choice of height scale at 1.2 m is a bit odd, and "Thickness" may be the wrong word to use here. Maybe depth relative to non-conformity and start with 0 there? Is Espirutu _10 m thick, or 10.9-9.6 m thick? Because it shows very little, the "Elemental Analysis" column is a bit frustrating. Why not show XRD results as in Figures 3 and 7? Or better yet in all figures with similar columns show alteration/mineralization reactions and products, which could reflect either XRD or elemental work (e.g. + calcite, + albite, -quartz: : :) if you have a mixed bag of analyses. Note typo in 4a "phyllosicilate". In

caption: Espirutu or Espiritu? –

We fixed spelling of Espiritu, changed scale on column, and added explanation of "thickness".

408/Figure 6: This cation has a lot of passive voice. Unclear to me if the nonconformity is cut by "throughgoing veins" (check typo there) or the veins cut the shear zone, but those scenarios have pretty different implications. – changes made to clarify text

414/Figure 7: Perhaps worth noting whether units are measured depth, relative to sea level, true vertical depth, etc. for clarity. – change made

425/Figure 8. Typo: "Granitiod" in A. "Pinkcoated" is not particularly helpful; perhaps ID the mineral, even if speculative, or just call them partially-mineralized, sealed, or stained fractures, etc. whatever the case may be. – change made

430/Figure 9. Same comment about measured depth for clarity. – change made

435/Figure 10. Cool plot. The "Depth" axis on the permeability column is perhaps redundant.

We have kept the depth axis for consistency between the two sub-figures.

444/Figure 11. Photomicrograph 2. What does "intensely weathered _60 m" mean? Consider argillic alteration or just argillization. Argillite is a rock type, not the product of argillic alteration. "Iron" capitalized in caption.

Correction to figure text and changes made to figure caption.

446/Figure 12: : :. "Phyllosilicate" (in figure) vs "weathering" in caption. What is the rationale for the circular flow path? – change made

---

## Author Comment (AC2) · 15 Jun 2020

1. Illustrations of the fractures. The figures largely focus on the petrographic features and I found that the fracturing mentioned in several places is not fully illustrated. This includes, for example, (1) the near-vertical to bedding-parallel bleached fractures (l.100), (2) basement-hosted slip surfaces (l.101), cm- to m's - displacement faults (l.115), slip surfaces with oblique to dip-slip slickenlines mm's to cm's thick (l.189). . ... I think the manuscript will be improved by illustrating these features. –

[Figure]

We have added labels on images where possible and have provided references to work done in MS theses.

2. Lateral variability. Nonconformity zones likely have properties that are spatially heterogeneous. This is indicated l.49-52: "Due to weathering, deformation, diagenesis and fluid-rock interactions, the nonconformity zone may be hydraulically heterogeneous at the mm to 10's m scales". The paper addresses the vertical variability, but it does not fully address the lateral variability, which is critical for fluid flow and can give insights into how core data can be extrapolated to large-scale application. The studied outcrops allow analysing this lateral variability. For example, rocks studied in section 2.2.2 are observed along a 4-km long section in Gallinas Canyon. I will recommend the authors to further this point. -change made - added detail

3. Geological settings. The description of the geological setting is very sparse. I recommend the authors to provide more information for each studied area. For example, the types and ages of the fractures and some general descriptions on the tectonic setting. Maybe providing the locations of the studied areas on geological maps could be useful.

We. have added a section on geologic setting with locations on maps.

4. Analogues. Two outcrops are examined: (1) a nonconformity between late Proterozoic Jacobsville Sandstone and early Proterozoic altered peridotite outcropping in Michigan and (2) the contact between Devonian to Mississippian carbonate and clastic rocks of the Espiritu Santo Formation deposited on the Proterozoic quartzofeldspathic and amphibolitic gneiss. Also, three cores are examined: (1) a core recovered from the Cambrian Lamotte Formation sandstone and sheared Proterozoic granitoids in south-central Nebraska, (2) from the Cambrian Mt. Simon Sandstone and Precambrian altered granitoid gneiss of the Grenville Front Tectonic Zone and (3) from a section of rocks of lower Cambrian Mt Simon Sandstone overlying a Precambrian layered intrusive complex. I think it is interesting to analyse these very different areas because it

provides an idea about the diversity of the nonconformity. However, the negative point is that it is not clear whether the studied areas are analogues to fluid injection sites or not. For example, in the introduction, the authors mentioned the mid-continent United States and the works by Murray (2015). This work concerns Oklahoma's underground injections, where target rocks are mostly carbonates from the Arbuckle sedimentary strata above a crystalline basement. My knowledge of the regional geology of the studied areas is limited, but I think it will be worthy to justify the choice of the studied areas and how these areas are relevant for fluid injection operations. I think the last conclusion point: "the contact . . .. should instead be evaluated on a site by site basis prior to injection of large fluid volumes" is critical in this regard.

We have added a statement regarding the choice of localities, and how they serve as analogues, the use of Oklahoma as one example of deep injection and associated earthquakes is meant to provide the reader with the research driver that deep injection into reservoirs above the nonconformity has led to earthquakes in crystalline basement –

5. Quantitative insights. The authors briefly describe permeability values measured for one core in section 2.2.4. This is important data and the results could be described further. Besides, I think it could be interesting to provide similar information from the other studied areas. Also, there is a mention of fractures with density decreasing with depth (l.189). I think this could be described quantitatively as well. For example, by providing a fracture density log. More generally I think providing further quantitative analysis will be welcome and will make this work more valuable.

We have added fracture data where possible and have provided references to work done in MS theses.

6. Numerical modelling. I think the last paragraph of the discussion on numerical modelling will be better in the result section. Also, I think that the authors should provide more information about their methodology, the origin of the permeability values,

the choice of the studied geometry and the boundary conditions.

We have moved the numerical modelling results to improve flow of the discussion.

Minor points 1. Mt. Simon or Mount Simon, both are used in the manuscript. – change made

2. l.204 "not the result of alteration due to weathering alone". I think this should be discussed further. – expanded explanation

3.The figure captions are often incomplete. – figure captions edited for completeness

4. Fig.2: What is A in the figure. There is no scale in B and C. – fixed typo

5. Fig. 3. There are no B, C and D in Fig. 3. I think the authors mean 4, 5 and 6. -change made Although there is no 6 in the figure?

6. Fig. 4: What is A, B and C. There is no scale in B. – 10 cm scale bar appears at bottom of core box photo 7. Fig. 6: B is not indicated in the caption.- added callout in caption

---

## Author Comment (AC3) · 15 Jun 2020

The Discussion presents some inferences about fluid circulation and the interpretations of structures and mineral deposits. As it stands some of this text seems speculative. The arguments should at least be bolstered by pointing to some of the extant structural diagenesis literature.

We have added citations to the fracture and diagenesis literature when discussing mineralogic changes.

[Figure]

Where the text describes 'fractures' and fracture mineralization, the descriptions could be more complete (and meaningful). More information could be provided on whether the fractures are 'opening mode' or faults. The use of the term 'vein' is unhelpful, particularly with respect to structures in the cover above the nonconformities. Mineral fill in fractures is common throughout sedimentary sequences (e.g. Laubach et al. 2019, Reviews of Geophysics) and such mineral deposits could provide evidence of the post depositional structural and fluid history of these zones. So a more meaningful description of these features could be useful. Note also that there are a number of published studies of fracture systems in basal Cambrian and in Ordovician sandstones of the midcontinent and other Laurentia cover rocks, and the fracture sets have a range of ages and origins.

We have provided specific descriptions of fracture types throughout and made call outs where possible in figures to identify the features.

Some statement as to how representative these outcrops are of the midcontinent non-conformity zones would be helpful.

We have added a statement on the midcontinent nonconformity study locations and their use as analogs (section 2).

I think I follow what you are saying here about the definition of the 'nonconformity zone', but perhaps the definition could use sharpening. Are you talking about some volume of rock near the nonconformity that is somehow altered from what it would be if the same rock was not near the nonconformity? Do you only mean rocks in the basement or could this include rocks above the nonconformity? Can you try to make the definition more explicit?

The nonconformity zone is the volume of rock adjacent to the nonconformity, in most cases it is altered, we have clarified this definition (53).

Where you mention 'the nonconformity' it might help reader if you remind them here that you mean 'the nonconformity in the US midcontinent region'. – Change made

The Introduction would be improved by adding an explicit claim here that could start with the statement 'here we show that: : :' Motivate the reader rather than just providing a list of what you did. – added

But are these overlying rocks mostly quartz-rich sandstones? Isn't the basal Cambrian sandstone pretty common? I see that you outline the geology you looked at in section 2.1. Do you discuss how representative these might be? –

We have added this information to Section 2 – Geologic setting.

Where in the Introduction do you alert the reader that you present modeling?

We have added reference to modeling into the introduction (77).

'detailed' is vague; can you replace this statement with a scale (or range of scales)? Or just omit, since the resolution level is implied by the instruments you used. – change made

Is there a reason for the order that you describe the localities? Same question for the listing in section 2.1. A representative selection?

Localities were grouped based on study and sampling sites being outcrop vs core. There is no specific order but the sites are a representative selection of the basement tectonic zones of US mid-continent. We have further addressed this in section 2.

How low is the porosity?

We have removed reference to porosity, as at this point in time it is only a qualitative observation from petrography.

if the fractures are bedding parallel as you say, it would be hard for them to extend into basement. Or do you mean the reduction spots are not in basement? – reworded sentence

[Figure]

Are these slip surfaces in basement subparallel to the bedding parallel 'fractures' in the cover. Are the cover fractures faults?

There is no evidence of slip observed in the bleached fractures in the Jacobsville unit. Evidence of slip was observed in the basement and align with the near-vertical bleached fractures in the cover.

By 'span the contact' do you mean the faults extend into the cover? – yes – reworded (184)

Something is awkward in the phrasing here. – reworded typo

Are you saying fault rock is only found in faults? Clarify text. – change made (line214)

Quartz lined and quartz-filled fractures are common in quartzose sandstones even distant from nonconformities. The mineral deposits may not necessarily represent mineralization 'events' since the fractures themselves are reactive surfaces (e.g. Lander and Laubach 2015, GSA Bulletin). – reworded and preceding text. What kind of 'fractures'; opening mode, or faults? Are there crosscutting relations here that provide evidence for the relative timing of these structures? Are you implying that the shear zone in the basement is somehow related to the fractures in the cover? (Wouldn't that be surprising?) – changes made (225>)

Is this the porosity range at the site you sampled? It seems a stretch to say that this is the range for the Mt Simon generally, since porosity ought to reflect thermal exposure/burial history and that could vary regionally. Clarify.- reworded space – change made

'multi-layered veins and/or fracture mineralization'; are these different things? – reworded

'porous'; but can you specify how porous? –

The porosity is qualitative based on petrographic observation at this time. No quantitative estimate of porosity was made and thus we have adjusted text.

'structural discontinuities' seems vague. We use this word because it encompasses all types of fractures, veins, faults across all study sites.

Is the thickness of the nonconformity zone specified at the outset of each description above? And how did you decide where the boundaries of the zones are? – added improved definition of the nonconformity zone

What is the opposite of 'in situ' mineral growth? – change made

Maybe put in a table? And refer to in description. – change made

The first paragraph of the Discussion seems vague and disorganized. Are these structures in the nonconformity zone' or in the basement or the cover? Are these only 'small faults' or are some of the fractures opening mode? - Discussion has been rewritten for clarity

The 'non fractured'; do you mean that these zones lack fractures in general, or that in areas where fractures happen to be absent, the host rock attributes might have these effects? – fractures are absent – reworded for clarity

'we note that: : :'; what is the basis for this inference? That there are porous rocks above the basement rocks? -removed statement

219-220 I don't see how it follows that the 'vein mineralogy' provides evidence for cross unconformity flow. Are you talking about mineral filled fractures in the basement or in the cover? Note that from mineral composition alone it can be challenging to find evidence for fluid flow (see for example, Denny et al.2020 GSA Bulletin). Maybe this point needs more development or the conclusion should be presented in a more nuanced way. – reworded

We see consistent mineralized fractures (vein) and cross-cutting relationships within the veins in the basement and sedimentary cover suggesting similar fluid rock interactions.

In the older rock mechanics literature there are examples of fracture systems in basement associated in typical midcontinent crystalline rocks that extend to depths of hundreds of meters and then abruptly stop; so zones of penetration of alteration could be much more than 5 m (and might be heterogeneous, if linked to deep seated fractures). Se references by Aubertin. – reworded

The sentence was reworded to reflect our direct observations (5 m) and those of previous workers Duffin.

'that impacts' or 'that would be expected to impact'? – change made

But are these the 'bed parallel' fractures? –

These are fractures that cut across the nonconformity at an angle, so no, not bed-parallel

What do you mean by 'deep circulation'? The basement rocks are not all that far from porous sedimentary rocks, which likely contain fluids.

Change made to avoid confusion. We were referring to 'deep' as in basement involved and not limited to circulation within the sedimentary rocks.

Where did you mention what the mechanical properties of these rocks is? Did you measure them, or is that an inference from the rock types? An example of mechanical properties inhibiting fracture in the setting you are concerned with is in Ellis et al. 2012, J. Geol. Soc. London. –

No mechanical properties were measured in this work. We have added a citation to support importance of rheology/mechanical change across boundary.

do you mean 'faults'? Structural discontinuities in this paper could include veins, joints, faults, and cataclastite zones.

259-264 Is this your claim? - reworded

Is this modeling work prefigured in the Introduction?

We have added reference to modeling work in introduction.

How representative are these various types you identify?

We provide a supplementary table with additional nonconformity sites (7 outcrop sites, 6 core total) at all sites the nonconformities fall into one of the three end-members. We have added text to the conclusion to reference supplementary table and representative nature of these end-members.

'Laubach' is the correct spelling. – change made

---

## Editor Comment (EC3) · Roger Soliva (Editor) · 22 Jun 2020

Dear Authors,

Thank you for the work and efforts made to revise the manuscript, which is now well improved, especially with the structure of the manuscript and the geological settings. Based on your reply, I now have more recommendations about the deformation structures and method for the models you present, that need to be adressed before the manuscript can be considered for publication.

[Figure]

1 - You mention that you have improved the descritption and labelling of the structures observed in the study sites (faults, veins etc...) but you did not clearly mentioned how and where you did these improvements. Please provide the tracking of these spêcific changes made and try to improve as much as possible these descriptions. You use the terms faults, fractures and veins, without clear definition of these terms (what you consider a fracture to be compared to a vein and a fault) but also the failure modes (slip, opening, hybrid...) you consider each time you describe them (if not clear, mention it). You add a new table with a synthesis of the structures you observed in the field, which could be useful to refer to in the results section (not only in the discussion).

2 - In your new method section, please provide explanations about the physics of the model used in Figure 12 (laws used for fluid transfer and pore pressure calculation) and the numerical method (finite element ?). Also add some references to other works using this model or at least this type of model exposing the full explanations.

Figure 6: Please, add the unit used for K in the caption text.

Photographs in general are very small, and particularly in Figure 6 and some images in Figure 8. It is then very difficult to see what you show.

I look forward receiving these revisions.

Best regards, Roger Soliva

---

## Author Response (AR2)

1 - You mention that you have improved the descritption and labelling of the structures observed in the study sites (faults, veins etc...) but you did not clearly mentioned how and where you did these improvements. Please provide the tracking of these spêcific changes made and try to improve as much as possible these descriptions. You use the terms faults, fractures and veins, without clear definition of these terms (what you consider a fracture to be compared to a vein and a fault) but also the failure modes (slip, opening, hybrid...) you consider each time you describe them (if not clear, mention it). You add a new table with a synthesis of the structures you observed in the field, which could be useful to refer to in the results section (not only in the discussion).

I apologize I must have not kept track of the changes in the previous version,       we have included fracture type in this version see results section where we include description of fracture types (lines: 185,220, 240-245; 264;274-275); in line 38 we added a definition of fracture zone in this manuscript, and expanded table 1 to included features associated with each nonconformity.

Figures 2,3,4,7,9 & 11 include descriptions of the deformation features observed

2 - In your new method section, please provide explanations about the physics of the model used in Figure 12 (laws used for fluid transfer and pore pressure calculation) and the numerical method (finite element ?). Also add some references to other works using this model or at least this type of model exposing the full explanations.

Revision made and detail added – lines 157-169

Figure 6: Please, add the unit used for K in the caption text.
Photographs in general are very small, and particularly in Figure 6 and some images

in Figure 8. It is then very difficult to see what you show.

Figures have been revised for size and definition of units in caption

[revised manuscript text omitted]

|---|---|---|---|
| 1201 / 3980 | Shear zone / La Motte Sandstone | Major: quartz
sample 3984 | ① Coarse grained rounded to sub-rounded, poorly sorted quartz cemented quartz sandstone, 2% pore space
PPL top, XPL bottom
sample 3987
0.5 mm |
| 1213 / 3990 | | Major: microcline, orthoclase, sanidine
Minor: chlorite-serpentine, bokite, glauconite, cameronite, vermiculite
sample 3995 | ② Medium grained, poorly sorted, sub-angular to sub-rounded quartz sandstone, clay and oxide cement, calcite vein fill, 5% pore space
0.5 mm
sample 3993 |
| 1225 / 4018 | | Top Precambrian | |
| 1225.3 / 4020 | Finely foliated granitic shear zone | Major: dolomite, sanidine, orthoclase
Minor: clinochlore, quartz
sample 4025 | ③ Weathered basement shear zone, quartz, Fe-oxide and calcite veins some porosity between ridge grains and neo-formed clays
2 mm
sample 4031.6 |
| 1228.3 / 4030 | Granite with veins | Major: chlorite-serpentine,, orthoclase, quartz
Minor: imgreite
sample 4032 | ④ Altered basement shear zone, chlorite lined shear planes, sericitiztation of k-spar grains
2 mm
sample 4032 |
| 1231.4 / 4040 | | Major: chlorite-serpentine, quartz, sanidine
Minor: carlinite, K,Na,Tl,V,U-Ox, vermiculite
sample 4036 | ⑤ Quartz, k-spar, plagioclase, with chlorite lined shear fractures, sericitization of k-spar grains at twinning lamella
2 mm
sample 4036 |

[revised manuscript text omitted]